# ESCRT-III-dependent adhesive and mechanical changes are triggered by a mechanism detecting alteration of septate junction integrity in *Drosophila* epithelial cells

**Thomas Esmangart de Bournonville[1,2], Mariusz K Jaglarz[3], Emeline Durel[1], Roland Le Borgne[1]***

[1]Univ Rennes, CNRS, IGDR (Institut de Génétique et Développement de Rennes) – UMR 6290, Rennes, France; [2]Global Health Institute, School of Life Science, Ecole Polytechnique Fédérale de Lausanne (EPFL), Lausanne, Switzerland; [3]Department of Developmental Biology and Invertebrate Morphology, Institute of Zoology and Biomedical Research, Jagiellonian University in Krakow, Krakow, Poland

**\*For correspondence:**
roland.leborgne@univ-rennes1.fr

**Competing interest:** The authors declare that no competing interests exist.

**Abstract** Barrier functions of proliferative epithelia are constantly challenged by mechanical and chemical constraints. How epithelia respond to and cope with disturbances of barrier functions to allow tissue integrity maintenance is poorly characterised. Cellular junctions play an important role in this process and intracellular traffic contribute to their homeostasis. Here, we reveal that, in *Drosophila* pupal *notum*, alteration of the bi- or tricellular septate junctions (SJs) triggers a mechanism with two prominent outcomes. On one hand, there is an increase in the levels of E-cadherin, F-actin, and non-muscle myosin II in the plane of adherens junctions. On the other hand, β-integrin/Vinculin-positive cell contacts are reinforced along the lateral and basal membranes. We found that the weakening of SJ integrity, caused by the depletion of bi- or tricellular SJ components, alters ESCRT-III/Vps32/Shrub distribution, reduces degradation and instead favours recycling of SJ components, an effect that extends to other recycled transmembrane protein cargoes including Crumbs, its effector β-Heavy Spectrin Karst, and β-integrin. We propose a mechanism by which epithelial cells, upon sensing alterations of the SJ, reroute the function of Shrub to adjust the balance of degradation/recycling of junctional cargoes and thereby compensate for barrier junction defects to maintain epithelial integrity.

## Editor's evaluation

The authors explore an interesting question: how do epithelial tissues respond to loss of barrier function in vivo? These important results break new ground in looking at the dynamic relationships between junctional complexes. The results of this convincing paper will be of interest to a broad audience of cell and developmental biologists.

## Introduction

Epithelia are key tissues of organisms, facing the outside and protecting the inner part of the organism against both physical and chemical injuries. Although the epithelial cells composing the tissue need to establish solid and resistant barriers, they remain highly plastic. Indeed, throughout development,

epithelial cells undergo profound changes in cell shape or cell–cell contacts during cell movements, divisions, cell intercalation, or extrusion (*Godard and Heisenberg, 2019*; *Matamoro-Vidal and Levayer, 2019*; *Perez-Vale and Peifer, 2020*; *Pinheiro and Bellaïche, 2018*). Most of these mechanisms imply junctional remodelling and rely on a set of molecular actors that form those junctions. The transmembrane protein E-cadherin (E-cad) connects to cellular cytoskeleton, made of proteins such as non-muscle myosin II (Myo-II) and filamentous actin (F-actin), through α- and β-catenins, and together they build up the adherens junction (AJ) (*Clarke and Martin, 2021*). AJs play the role of a mechanical barrier in the tissue, ensuring that cells are closely packed and resistant to physical stress (*Charras and Yap, 2018*).

Basal to AJs, in *Drosophila* epithelia, a second type of intercellular junctions are septate junctions (SJs), appearing as a ladder-like structure at the electron microscope resolution. SJs provide a paracellular diffusion barrier to solutes, similar to that of tight junctions in vertebrates (*Genova and Fehon, 2003*; *Ward et al., 1998*). In addition to the occludens barrier function, SJs also exert a structural role including cell adhesion, cell polarity, and cell shape regulation (*Laprise et al., 2009*; *Rice et al., 2021*). The cytosolic polarity regulators Scribble, Discs Large, and Lethal Giant Larvae are SJ resident proteins (*Izumi and Furuse, 2014*; *Rice et al., 2021*), but not per se SJ components. The so-called 'core SJ complex' is composed of cell adhesion proteins including Neurexin-IV (Nrx-IV) and Neuroglian (*Baumgartner et al., 1996*; *Genova and Fehon, 2003*), Claudin-like family of proteins (*Behr et al., 2003*; *Nelson et al., 2010*; *Wu et al., 2007*), and cytosolic proteins such as Coracle (Cora) (*Lamb et al., 1998*) and Varicose (*Wu et al., 2007*). At the meeting point of three cells within an epithelium, a specialised domain called a tricellular junction (TCJ) arises, and to date three proteins have been described as enriched at the SJ level: Gliotactin (Gli) (*Schulte et al., 2003*), Anakonda (Aka; also known as Bark Beetle) (*Byri et al., 2015*; *Hildebrandt et al., 2015*), and the myelin proteolipid protein family member M6 (*Dunn et al., 2018*). We and others have recently described an intricate interplay in which both Aka and M6 are required to recruit and stabilise themselves at the TCJ, while Gli is needed to stabilise them both at the TCJ (*Esmangart de Bournonville and Le Borgne, 2020*; *Wittek et al., 2020*). Moreover, we have shown that TCJ proteins are required to ensure the anchoring of SJ proteins at the three-cell contact, also called a vertex, and, in turn, vertex-specific enrichment and restriction of TCJ proteins are linked to SJ integrity (*Esmangart de Bournonville and Le Borgne, 2020*).

As described above for AJs, SJs must also be highly plastic to cope with a high rate of cell division, tissue growth, cell intercalation, or delamination, while maintaining the integrity of the permeability barrier. Our previous work contributed to show that SJs are stable complexes, exhibiting a turnover rate of 90 min. SJ components are delivered and assembled apically, just basal to AJs, and continue to be progressively dragged basally in a treadmill-like manner (*Daniel et al., 2018*). At the basal SJ belt, SJ components are thought to be disassembled, internalised, and recycled apically to form new SJs or to be degraded. Several studies have revealed that intracellular trafficking actors, such as Rab11 (*Dong et al., 2014*), the retromer, and the endosomal sorting complexes required for transport (ESCRT)-III component Shrub, are key regulators of SJ establishment and integrity (*Pannen et al., 2020*). Retromer is implicated in the retrieval of cargoes from endosomes while ESCRT-III regulates ubiquitin-dependent degradation of transmembrane cargoes. In addition, the Ly6-like proteins Crooked, Coiled, Crimpled (*Nilton et al., 2010*), and Boudin (*Hijazi et al., 2009*; *Tempesta et al., 2017*), four SJ accessory proteins required for SJ assembly, have been reported to regulate the endocytic trafficking of Nrx-IV and Claudin-like Kune-Kune.

Despite the fact that SJs have been extensively studied for the past decades, it only recently emerged that they might be involved in additional mechanisms beyond their initially described filtering actions (*Rice et al., 2021*). For instance, a striking feature of *Drosophila* embryo SJ mutants is the appearance of a wavy trachea associated with defects in SJ-mediated endocytic trafficking. Other morphogenetic defects include diminished and deformed salivary glands, head involution, and dorsal closure defects. SJ proteins also regulate the rate of division of intestinal stem cells (*Resnik-Docampo et al., 2021*; *Resnik-Docampo et al., 2017*), as well as hemocyte lineage differentiation via interactions with the Hippo pathway (*Khadilkar and Tanentzapf, 2019*; *Khadilkar et al., 2017*). Another intriguing feature is the confirmation of the role of SJ components in wound healing (*Carvalho et al., 2018*). Indeed, lack of different SJ components impairs the formation of actomyosin cables, which are regulated by AJs and under normal conditions ensure the proper healing of the tissue. Hence,

the studies cited revealed that SJ proteins can impact mechanical properties of the tissue, calling for a deeper understanding of the impact that the loss of SJ integrity has on general mature tissue homeostasis.

We recently reported that defects at tricellular SJs (tSJs) are accompanied by bicellular SJs (bSJs) defects. Indeed, restriction of tSJ components at the vertex is dependent on bSJ integrity. Conversely, loss of tSJ components causes considerable membrane deformation and the loss of bSJs abutting the vertex (*Esmangart de Bournonville and Le Borgne, 2020*). However, and surprisingly, under these conditions, cells remain within the epithelial layer and do not delaminate. Also, in embryonic and larval epithelia lacking tSJs, bSJs assemble initially, but degenerate later in development (*Byri et al., 2015*; *Hildebrandt et al., 2015*). In this paper, we investigate how cell adhesion is modulated and allows epithelial integrity to be maintained following disruption of the integrity of SJs. We use the *Drosophila* pupal *notum* as a model of mature epithelium with established and functional mechanical and paracellular diffusion barrier functions. This tissue lends itself to quantitative imaging in which we can easily dissect the mechanics and genetics of epithelia.

## Results

### Disruption of tSJ and bSJ integrity alters the distribution of AJ components

We have previously described that NrxIV-labelled bSJs no longer terminate at vertices when TCJ components are lost (*Esmangart de Bournonville and Le Borgne, 2020*). Here, we carried out the following morphometric experiment on RNAi-treated tissue, allowing us to compare wild-type (WT) and *aka* mutant tissues. At the electron microscopy resolution, analysing thin sections parallel to the plane of the epithelium, we report that, depletion of Aka induces weaknesses in tissue integrity manifested by the appearance of sizeable intercellular holes in the plane of SJs (*Figure 1A–A″*). These observations are reminiscent to the paracellular cavities observed in embryos lacking Aka or Gli, interpreted as being due to a loss of cell–cell adhesion (*Byri et al., 2015*; *Hildebrandt et al., 2015*; *Schulte et al., 2003*). To investigate whether this morphological defect affects overall epithelial integrity, we studied the relationship between tSJs and AJs using clonal mosaic cell approach. In this figure (*Figure 1*) and following figures, clone boundaries in the AJ plane are indicated by yellow dashed lines and have been determined as described in *Figure 1—figure supplement 1A–A′*. We measured a 2-fold enrichment of *Drosophila* E-cad tagged with GFP (E-cad::GFP) at tAJs and 1.5-fold enrichment at bAJs in $aka^{L200}$ mutant cells (*Figure 1B–B′*). The increased signal of E-cad::GFP was accompanied by an enrichment of junctional Myo-II tagged with GFP (Myo-II::GFP) both at tAJs (1.7-fold enrichment) and bAJs (1.8-fold enrichment; *Figure 1C–C′*). The junctional and medial pools of Myo-II act in synergy with forces exerted by the medial–apical meshwork transmitted onto the junctional pool (*Lecuit and Yap, 2015*). The medial–apical network was also stronger in $aka^{L200}$ cells than in WT cells (1.5-fold enrichment; *Figure 1C–C′*). In addition, we probed F-actin and determined that loss of Aka resulted in a 1.9-fold and 2.5-fold increase in staining at bAJs and tAJs, respectively (*Figure 1D–D′*). We observed similar results upon loss of Gli, resulting in a 2-fold enrichment of E-cad at both bi- and tricellular junctions (*Figure 1—figure supplement 1B–B′*), suggesting that loss of tSJ components is responsible for the observed defects. Next, using a hypomorphic allele of the transmembrane bSJ protein Nervana 2 (Nrv2), we found that E-cad::GFP (*Figure 1E–E′*) and Myo-II::GFP (*Figure 1F–F′*) were enriched at both bAJs (E-cad::GFP 2.5-fold enrichment, Myo-II::GFP 2-fold enrichment), tAJs (E-cad::GFP 2.3-fold enrichment, Myo-II::GFP 2.3-fold enrichment), and medial network (Myo-II::GFP 2.3-fold enrichment) in $nrv2^{k13315}$ cells compared with WT cells. E-cad::GFP enrichment was also observed upon loss of GPI-anchored bSJ protein Coiled (cold) at BCJ (2.3-fold enrichment) and vertices (2.2-fold enrichment) (*Figure 1G–G′*). Those results indicate that alteration of the SJ resulted in increased levels of E-cad in the plane of AJ and thus raises the possibility of concomitant changes in epithelial cell adhesive and mechanical properties, which we have subsequently studied.

### The loss of Anakonda alters the adhesive and the mechanical epithelial properties

Because AJs are sites of mechanical force transduction, we hypothesised that the higher levels of E-cad and Myo-II modify the mechanical properties of the tissue. To assess it, we first tested if Myo-II

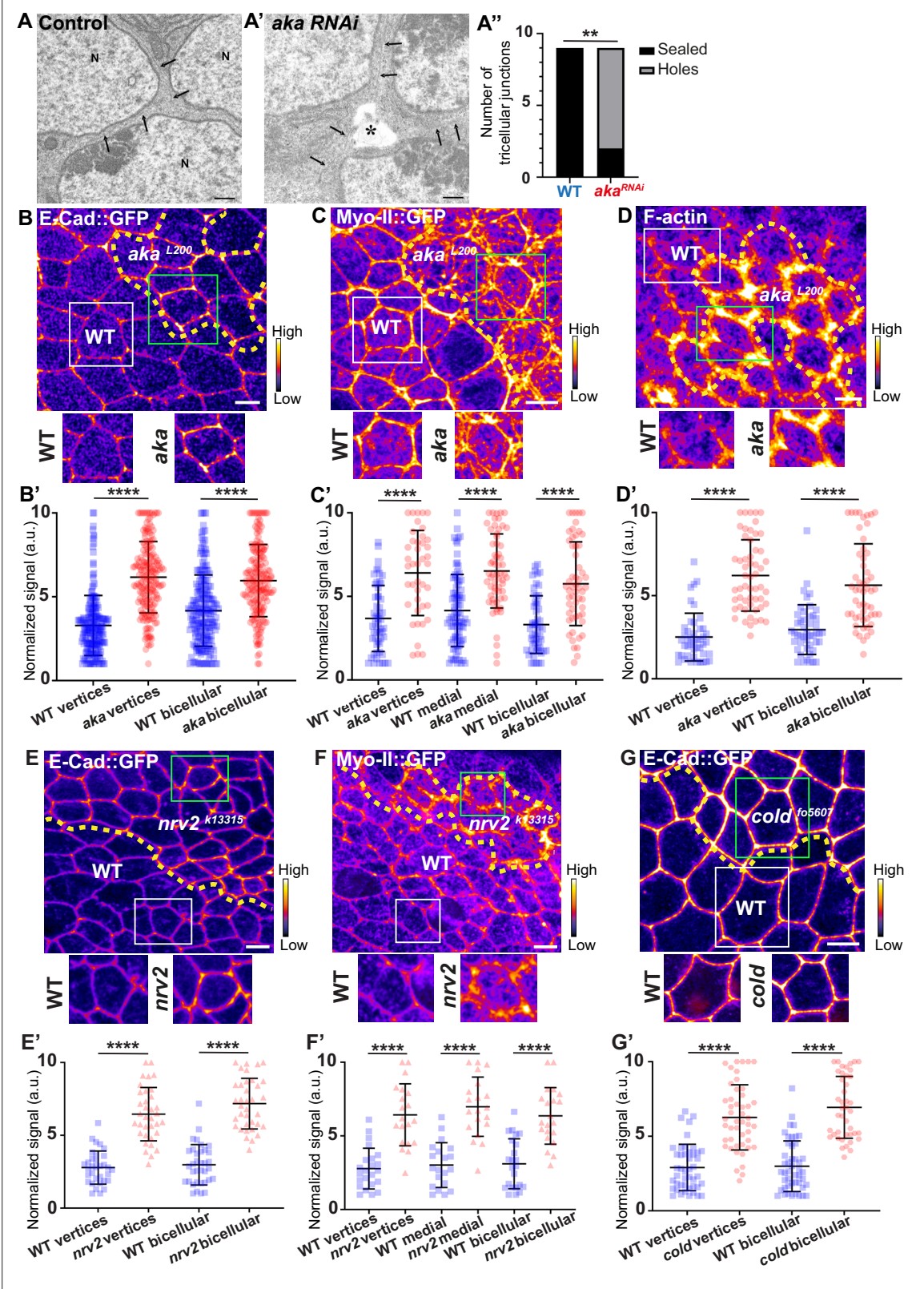

**Figure 1.** Consequence of loss of Anakonda on tricellular septate junction morphology and adherens junction components. Transmission electron microscopy of wild-type (**A**) and aka RNAi (**A'**) pupal *notum*. Note that Aka-depleted cells are separated by a large intercellular gap (asterisk) at the tricellular junction at the level of the nucleus. N: cell nucleus; arrows: cell membranes. (**A''**) Histogram representing the number of tricellular junctions being sealed (black) or not (grey) (n=9 and n=9 in WT (wild-type) and pnr>Aka-RNAi conditions respectively; n=3 pupae for each conditions). (**B–B'**,

*Figure 1 continued on next page*

*Figure 1 continued*

**C–C', D–D', E–E', F–F', and G–G'**) Localisation of E-cad::GFP (**B**, **E**, and **G**, fire colour), Myo-II::GFP (**C** and **F**, fire colour) or F-actin (**D**, phalloidin, fire colour) in wild-type, *aka*^L200^, *nrv2*^k13315^, and *cold*^f05607^ cells. Wild-type cells are separated from mutant cells by the dashed yellow line. (**B'**) Plot of the standardised E-cad::GFP signal at bicellular junctions and vertices in wild-type (blue squares) and *aka*^L200^ cells (red circles) (n=201 and 193 vertices and n=208 and 188 bicellular junctions for wild-type and *aka*^L200^ respectively; >5 pupae for each condition). (**C'**) Plot of the standardised Myo-II::GFP signal at bicellular junctions, vertices as well as medial network in wild-type (blue squares) and *aka*^L200^ cells (red circles) (n=54 and 42 vertices and n=84 and 61 cellular medial networks and n=55 and 56 bicellular junctions for wild-type and *aka*^L200^, respectively; n=5 pupae for each condition). (**D'**) Plot of the standardised F-actin signal at bicellular junctions and vertices in wild-type (blue squares) and *aka*^L200^ cells (red circles) (n=45 and 55 vertices and n=47 and 54 bicellular junctions for wild-type and *aka*^L200^, respectively; n=5 pupae for each condition). (**E'**) Plot of the standardised E-cad::GFP signal at tricellular and bicellular junctions in wild-type (blue squares) and *nrv2*^k13315^ cells (red triangles) (n=33 and 35 vertices and n=35 and 36 bicellular junctions for wild-type and *nrv2*^k13315^, respectively; 2 pupae for each condition). (**F'**) Plot of the standardised Myo-II::GFP signal at bicellular junctions, vertices as well as medial network in wild-type (blue squares) and *nrv2*^k13315^ cells (red triangles) (n=23 and 20 vertices and n=20 cellular medial networks and n=23 and 21 bicellular junctions for wild-type and *nrv2*^k13315^, respectively; n=2 pupae for each condition). (**G'**) Plot of the standardised E-cad::GFP signal at tricellular and bicellular junctions in wild-type (blue squares) and *cold*^f05607^ cells (red hexagons) (n=45 and 47 vertices and n=46 and 42 bicellular junctions for wild-type and *cold*^f05607^, respectively; 3 pupae for each condition). Bars show mean ± SD, **p<0.01, ****p<0.0001, Mann–Whitney test. A calibration bar shows LUT for grey value range. The scale bars represent 500 nm for panels **A–A'** and 5 µm for panels **B–G**. White squares represent close-up of WT and green squares of mutant situations for panels **B**, **C**, **D**, **E**, **F**, and **G**.

The online version of this article includes the following figure supplement(s) for figure 1:

**Figure supplement 1.** Consequence of loss of Nervana2 and Anakonda on E-cad and Myo-II localisation, and on cell–cell junction mechanical properties.

was activated in *aka* mutant context, by using an antibody against phosphorylated Myo-II (p-Myo-II), and we observed an enrichment in *aka*^L200^ cells compared with WT cells (*Figure 2A–A'*). The enrichment was of 1.6-fold at tAJs and bAJs and of 1.8-fold at the medial–apical network (*Figure 2A–A'*). Next, we probed junctional tension using two-photon laser-based nanoablation in the plane of the AJ labelled with E-cad::GFP (*Figure 1—figure supplement 1C–D*). Intriguingly, no significant differences in recoil velocities were observed upon ablation of WT cells versus *aka*^L200^ mutant junctions (mean = 0.19 ± 0.08 µm/s in WT vs mean = 0.20 ± 0.07 µm/s in *aka*^L200^) or *nrv2*^k13315^ cells (mean = 0.15 ± 0.07 µm/s in WT vs mean = 0.16 ± 0.08 µm/s in *nrv2*^k13315^) (*Figure 1—figure supplement 1D*). While recoil velocities indicated that there was no change in in-plane membrane tension upon loss of Aka, we noticed that the cell area of *aka*^L200^ cells was slightly reduced by 12% compared to WT (*Figure 1—figure supplement 1E*). This prompted us to analyse the length of the new adhesive interface formed during cell cytokinesis. Indeed, when a cell divides and forms its new cell–cell adhesive interface at the AJ level, the length of the new junction is determined by various factors: the force balance between the cells' autonomous strength in the actomyosin contractile ring, the cells' non-autonomous response of neighbouring cells that recruit contractile Myo-II at the edges to impose the geometry/length of the new interface, and the strength of intercellular adhesion defining the threshold of disengagement (*Founounou et al., 2013*; *Guillot and Lecuit, 2013*; *Herszterg et al., 2013*; *Morais-de-Sá and Sunkel, 2013*). Notably, E-cad overexpression was reported to delay junction disengagement leading to a shorter interface in early embryos (*Guillot and Lecuit, 2013*). First, we observed that when a WT cell divides between one WT and one *aka*^L200^ cell, the Myo-II::GFP signal was higher during the formation of and at the future vertex formed at the interface between WT and *aka*^L200^ cell, where there is no Aka (*Figure 2B–D*; white arrow) compared to the WT interface (*Figure 2B–D*; green arrow). While this phenomenon can be observed in WT conditions, the proportion of asymmetric enrichment of Myo-II::GFP was much higher in *aka*^L200^ conditions (*Figure 2D*). Then, we confirmed that WT cells established a long E-cad adhesive interface upon completion of cytokinesis, with few fluctuations in length and across time over the 30 min after the onset of anaphase (*Figure 2E and G*) as expected from *Founounou et al., 2013*; *Herszterg et al., 2013*. In contrast, *aka*^L200^ cells showed a reduction in this junctional length, as highlighted in some extreme cases of shrinkage (*Figure 2F, G*). This change in the new cell–cell interface length observed in *aka*^L200^ cells started to be significant approximately 10 min after the onset of anaphase (*Figure 2G*), suggesting fewer resisting forces from neighbours and/or increased constriction from the dividing cell.

To further explore defects in adhesive properties and mechanical tension caused upon loss of Aka, we examined the localisation of Vinculin (Vinc), an F-actin binding partner recruited at junctions in a tension-dependent manner (*Kale et al., 2018*; *le Duc et al., 2010*). We observed higher levels of GFP-tagged Vinc (Vinc::GFP) at tAJs (2-fold enrichment) and bAJs (1.75-fold enrichment) in *aka*^L200^ cells

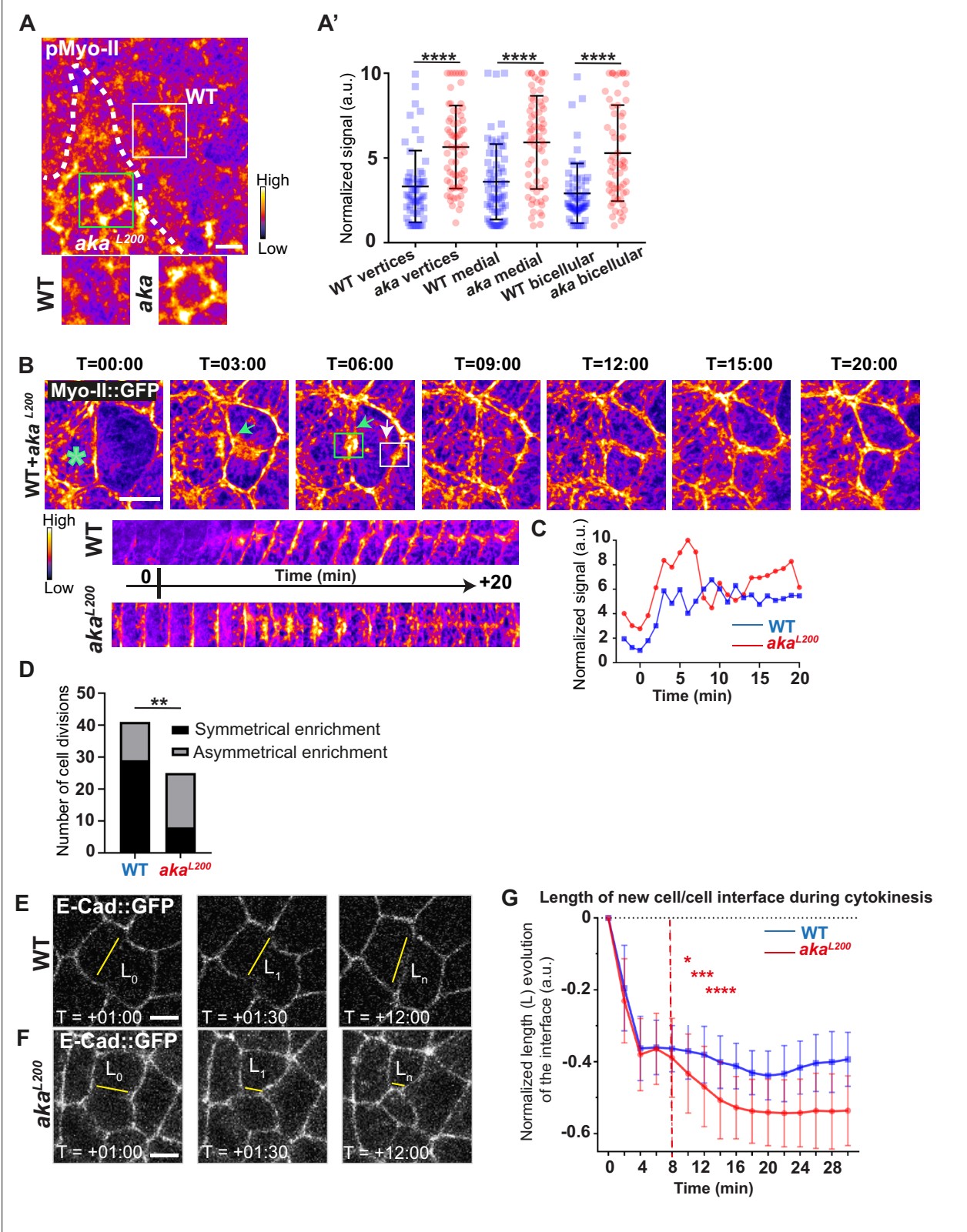

**Figure 2.** Loss of Anakonda promotes Myo-II activation and mechanical disturbances at adherens junction level during interphase and cytokinesis. (**A**) Shows example of a *notum* stained with anti-phospho-Myo-II (pMyo-II; fire colour), between 16 hr and 19 hr after puparium formation (APF), after heat-shock to induce clone of wild-type (WT) and mutant cells for Aka. (**A'**) Plot of the standardised pMyo-II signal at tri- and bicellular junctions as well as medial network in WT (blue squares) and *aka^L200* cells (red circles) (n=57 and 67 vertices, n=65 and 66 cellular medial networks and n=62 and 61

*Figure 2 continued on next page*

*Figure 2 continued*

bicellular junctions for WT and *aka^L200^*, respectively; n>5 pupae for each condition). (**B**) Cytokinesis of a WT cell expressing Myo-II::GFP between 16 hr and 19 hr APF, after heat-shock to induce clone of WT and mutant cells for Aka. Representation of a WT cell cytokinesis with recruitment of a higher amount of Myo-II::GFP at the contact with *aka^L200^* cell (marked by the green asterisk, green arrow for Myo-II::GFP signal) compared to the WT one (white arrow). Myo-II::GFP recruitment is asymmetrical in terms of Myo-II::GFP signal intensity. Kymograph represents the asymmetric enrichment of Myo-II::GFP of the WT and *aka^L200^* newly formed vertices depicted above. (**C**) Plot representing the Myo-II::GFP signal during cytokinesis at the WT (blue line) and *aka^L200^* (red line) newly formed vertices depicted in C. Time is min:s with t=0 corresponding to the anaphase onset. (**D**) Histogram representing the number of cells displaying symmetrical (black) or asymmetrical (dark grey) Myo-II::GFP recruitment during cytokinesis of WT with WT neighbours and of WT with one WT and one *aka^L200^* neighbours (n=29 and n=12; n=8 and n=17 for symmetrical and asymmetrical enrichment in WT and *aka^L200^* conditions respectively; n>5 pupae for each conditions). (**E–F**) Cytokinesis of *notum* cells expressing E-cad::GFP at 16 hr APF, after heat-shock to induce clone of WT (**E**) and *aka^L200^* mutant cells (**F**). Time is min:s with t=0 corresponding to the anaphase onset. L represents the length of the new cell/cell interface. (**G**) Plot of the mean length interface at each corresponding time points. WT situation is represented by blue squares and *aka^L200^* situation is represented by red circles. Bars show mean ± SD, *p<0.05, **p<0.005, ***p=0.0001, ****p<0.0001, unpaired t test and Mann–Whitney test for panels **A'**, Fisher t test for panel **D**, and Multiple t test for panel **G**. A calibration bar shows LUT for grey value range. The scale bars represent 5 µm. White square represents close-up of WT and green square of *aka^L200^* situations for panel **A**.

The online version of this article includes the following figure supplement(s) for figure 2:

**Figure supplement 1.** Loss of Anakonda leads to enrichment of Vinculin, Karst, and Ajuba at bi- and tricellular junctions.

compared with WT cells (***Figure 2—figure supplement 1A, C***). Strikingly, upon loss of Aka, Vinc::GFP was found enriched not only at the AJ level but also at the basal part of mutant cells (***Figure 2—figure supplement 1A', B***), raising the possibility of a reorganisation of the F-actin-anchoring point to the membrane associated with increased tension at these localisations (see below). We also found that the F-actin crosslinker, Karst, was enriched at the AJ level at bAJs (1.4-fold enrichment), at tAJs (1.6-fold enrichment), and at the apical–medial part of the cell (1.2-fold enrichment; ***Figure 2—figure supplement 1D-D'***). Then, we investigated the localisation of the Hippo/YAP partner Ajuba (Jub), known to be increased at AJ upon increased tension in *Drosophila* wing discs (***Rauskolb et al., 2014***). We observed an increase of GFP-tagged Jub (Jub::GFP) marking at tAJs (1.4-fold enrichment) and at bAJs (1.75-fold enrichment) (***Figure 2—figure supplement 1E–E'***). Collectively, these results suggest that the loss of Aka and concomitant disruption of SJ integrity increase apical tension and/or adhesive properties in epithelial cells. The mechanisms through which alteration of SJ components impacts AJ were then investigated.

## SJ alterations are associated with ESCRT complex defects

Several studies have revealed that the establishment and integrity of bSJs rely on intracellular traffic (***Nilton et al., 2010***; ***Pannen et al., 2020***; ***Tiklová et al., 2010***). Among them, Vps35 subcellular localisation is regulated by Shrub, which is itself needed to ensure correct bSJ protein delivery at the plasma membrane. In the pupal epithelium, loss of Shrub causes loss of ATP-α::GFP signal, indicative of an interplay between SJs and endosomal sorting machinery (***Pannen et al., 2020***). Upon loss of Aka, bSJs are no longer connected to vertices and exhibit membrane deformation with increased levels of bSJ components (***Esmangart de Bournonville and Le Borgne, 2020***). The higher level of bSJ components could result from an increased delivery of newly synthesised proteins, reduced endocytosis, and/or increased recycling of bSJ proteins. We hypothesise that defects in SJ integrity might feedback on the endocytosis recycling of bSJ proteins, to compensate for SJ defects. To probe for possible membrane traffic alterations, we investigated the ESCRT complex by examining the multivesicular body (MVB) marker, the ESCRT-0 component hepatocyte-growth-factor-regulated tyrosine kinase substrate (HRS)/Vps27 and Shrub/Vps32 endogenously tagged with GFP (Shrub::GFP). We performed a knock-down of Cora using RNAi. In the control portion of the tissue, which is the part of the *notum* where the Pannier (Pnr) is not expressed (***Figure 3A–B'***), we observed that HRS and Shrub::GFP appeared as small punctate structures that partially colocalise (white structures; ***Figure 3A–B'***). Strikingly, silencing of Cora induced the formation of enlarged Shrub::GFP-positive structures, more and larger HRS-positive compartments (***Figure 3C–D'***), together with bSJ integrity alteration (***Figure 3C'***). The enlarged Shrub::GFP-positive structures did not colocalise with HRS punctae (***Figure 3C–D'***). We also detected larger and brighter HRS-positive structures, both in *aka^L200^* (***Figure 3—figure supplement 1A, B***) and in *nrv2^k13315^* cells (***Figure 3—figure supplement 1C, D***).

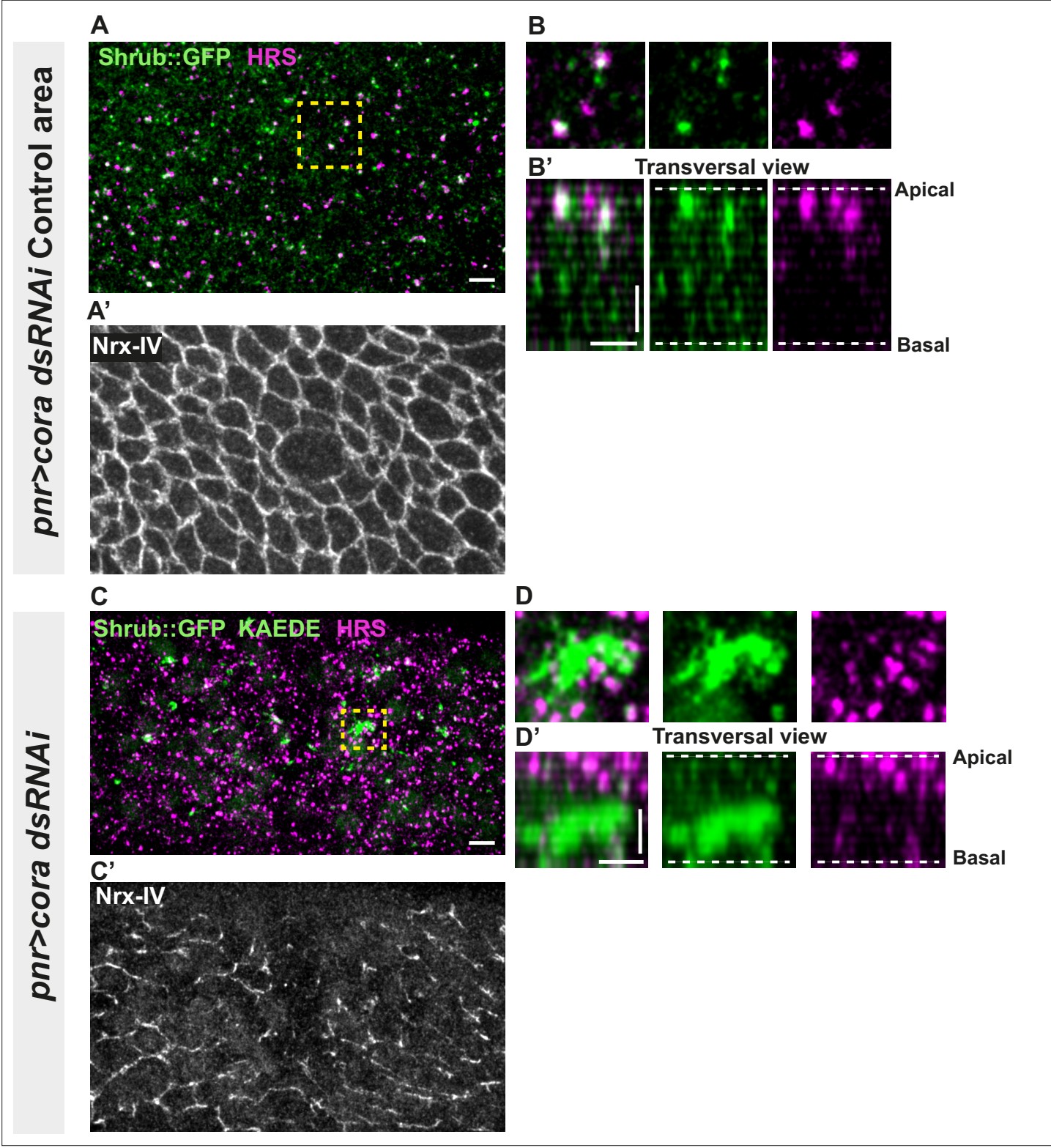

**Figure 3.** Septate junction (SJ) defects are associated with increased number of HRS- and ESCRT III protein Shrub-positive structures. (**A–B′** and **C–D′**) Localisation of Shrub::GFP+GFP antibody (green), KAEDE (**C–D′**) in cells marked by Nrx-IV (anti-Nrx-IV, grey) and HRS (anti-HRS, magenta) in wild-type and cells expressing UAS::*cora*-RNAi together with UAS::KAEDE under *pnr*-Gal4 control. (**A–B′**) Localisation of Shrub::GFP+GFP antibody and HRS in a wild-type area of a tissue expressing UAS::*cora*-RNAi and UAS::KAEDE under *pnr*-Gal4 control (KAEDE negative) and regular Nrx-IV signal in (**A′**) in a planar view (**A**, **A′**, and **B**) or in a transversal view (**B′**). Yellow dashed square shows (**B** and **B′**) magnification of wild-type cell with colocalisation

*Figure 3 continued on next page*

*Figure 3 continued*

between Shrub::GFP and HRS at SJ level shown by Nrx-IV. (**C–D'**) Localisation of Shrub::GFP+GFP antibody and HRS in cells expressing UAS::*cora*-RNAi and UAS::KAEDE under *pnr*-Gal4 control (KAEDE-positive) cells and Nrx-IV reduced signal in (**C'**) in a planar view (**C, C'**, and **D**) or in a transversal view (**D'**). Yellow dashed squares show (**D–D'**) magnification of aggregates of Shrub::GFP surrounded by HRS staining. The scale bar represents 5 µm (**A** and **C**) and 3 µm in (**B'** and **D'**). Dashed white lines highlight apical and basal limits of the *notum*.

The online version of this article includes the following figure supplement(s) for figure 3:

**Figure supplement 1.** Loss of Anakonda or Nervana 2 triggers increased number of HRS-positives vesicles.

Because the ESCRT complex is involved in controlling the degradation of poly-ubiquitinylated cargoes (*Cullen and Steinberg, 2018*), we then asked whether the excess of Shrub-positive enlarged structures was due to a change in Shrub degradation activity. A way to probe putative defects in ESCRT function is to monitor the amount of poly-ubiquitinylated proteins targeted for degradation (*Cullen and Steinberg, 2018*). First, we used an anti-FK2 antibody, a monoclonal antibody targeting poly-ubiquitinylated proteins, in a Shrub RNAi context and confirmed that depletion of Shrub led to both appearance of poly-ubiquitinylated proteins aggregates and SJ alterations as observed by the inhomogeneous Nrx-IV signal (*Figure 4A*). Then, using the Cora-RNAi approach again, we observed Shrub::GFP and poly-ubiquitinylated proteins FK2 as small punctate compartments in the control portion of the tissue (*Figure 4B*). In striking contrast, in the Cora-depleted domain, Shrub::GFP and anti-FK2 labelled large structures (*Figure 4C–C"*). Shrub::GFP-positive structures were closely juxtaposed and/or partially colocalised with FK2 (*Figure 4C'–C"*). Similar observations were made upon knock-down of Nrx-IV (*Figure 4D–D"*), as well as in *aka*^L200 cells (*Figure 4E*). Hence, mutants with disrupted SJ integrity display features of a dysfunctional ESCRT-III-dependent degradation pathway, somewhat reminiscent of a *shrub* loss of function. Despite these apparent similarities, we noticed that, in contrast to Shrub depletion (*Bruelle et al., 2023*), NrxIV did not accumulate in enlarged intracellular compartments upon Cora depletion (*Figure 4C and C"'*). In other words, the accumulation of Shrub::GFP in enlarged compartments seen upon Cora depletion is not functionally equivalent to the loss of Shrub. We propose that it is the Shrub activity that is being modified upon SJ alteration, preventing SJ component degradation in favour of SJ component recycling. In support of this proposal of increased recycling, loss of TCJ components was shown to cause membrane deformations enriched in SJ components (*Esmangart de Bournonville and Le Borgne, 2020*). The next question was whether deregulation of Shrub activity by SJ component depletion could affect adhesive properties and cell mechanics.

## Loss of tSJ or bSJ components impact Crumbs localisation and triggers assembly of focal adhesion contacts

In *Drosophila* trachea, loss of Shrub has been reported to affect the localisation of bSJ components, such as Kune-Kune, impairing the paracellular diffusion barrier on one hand and Crb activity on the other (*Dong et al., 2014*). Loss of Shrub results in an elongated sinusoidal tube phenotype which was shown to be caused by mislocalised Crb activity. Indeed, in *shrb*^4 clones, instead of being restricted to the junctional domain, Crb is present in ESCRT-0-positive endosomal compartments causing Crb activity in endosomes (*Dong et al., 2014*). In this study, the authors raised the possibility that the defect of bSJ caused by loss of Shrub might also contribute to an excess of Crb activity, a possibility that we then tested. As a control, we monitored the localisation of SJ protein Kune Kune (Kune) and Crb using an anti-Crb antibody targeting its N-terminal extracellular domain (anti-Crb). We showed a colocalisation in small vesicles at the basal level of the cell (white vesicles; *Figure 5—figure supplement 1A–A"*), suggesting that Kune and Crb traffic together. Upon knock-down of Shrub via RNAi, we observed defects of Kune and Crb characterised by enrichment of Crb and Kune in basal aggregates (*Figure 5—figure supplement 1B–B"'*). The apparent similarities between depletion of Shrub and that of b/tSJ components on FK2 and HRS raised the question whether the loss of Aka could result in defective Crumbs localisation. To investigate this possibility, we monitored Crb localisation in tSJ defects situation using Crb tagged with a GFP in its extracellular domain (Crb::GFP) or an anti-Crb antibody in *aka*^L200 context. Crb signal was detected both at junctional and medial apical parts of WT cells (*Figure 5A and C*). Strikingly, in *aka*^L200 and in bSJ defective *nrv2*^k13315 cells, the apical–medial Crb signal was increased (*Figure 5A–C'* and *Figure 5—figure supplement 2A–C*). Concerning

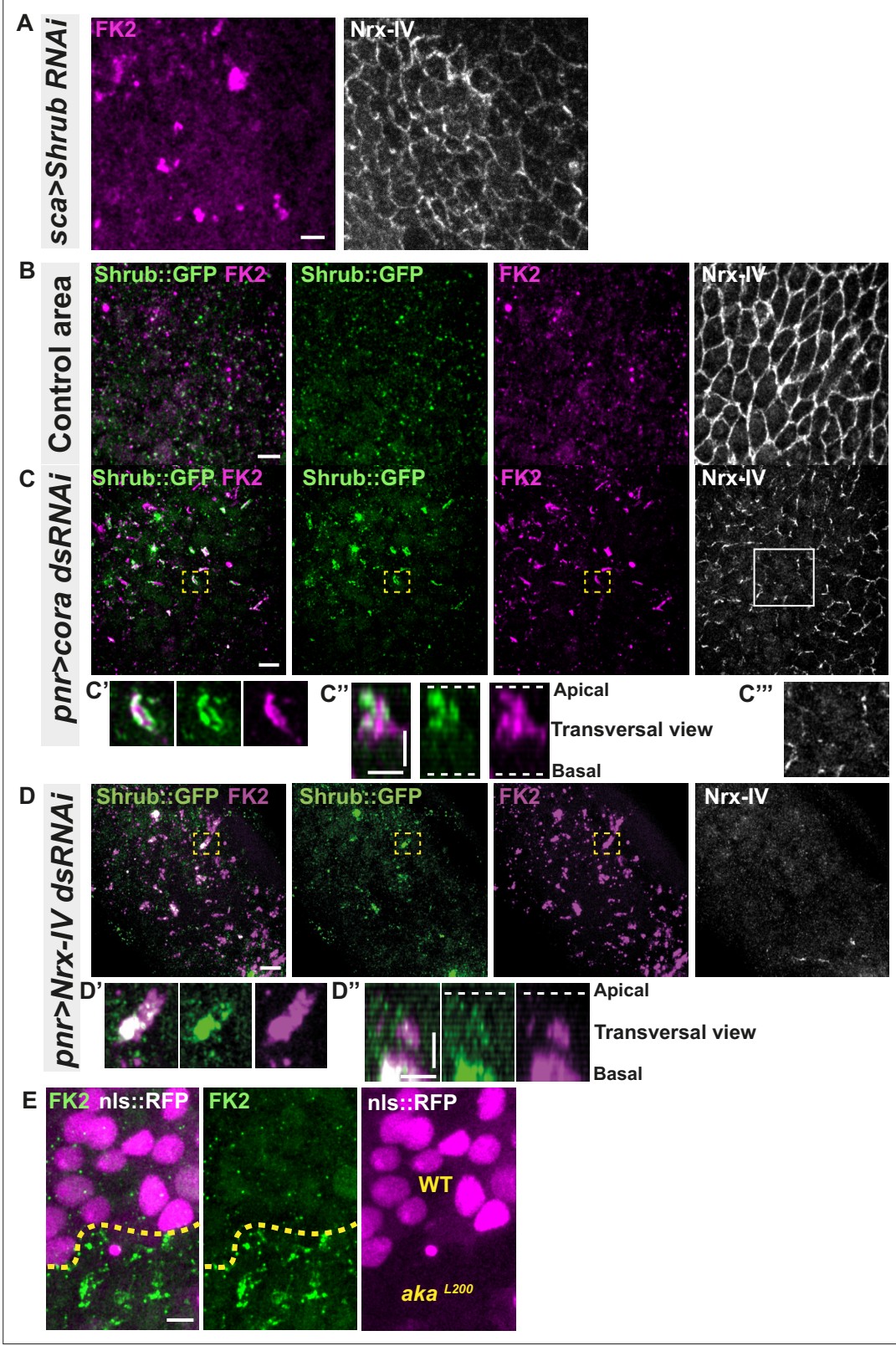

**Figure 4.** Septate junction defects leads to the enlargement of ESCRT III protein Shrub- and ubiquitinylated proteins-positives structures. (**A**) Localisation of FK2 (anti-ubiquitinylated proteins, magenta) in cells marked by Nrx-IV (anti-Nrx-IV, white) expressing UAS::*Shrub*-RNAi under *sca*-Gal4 control. (**B–C'''**) Localisation of Shrb::GFP+anti-GFP antibody and FK2 in a wild-type area (regular Nrx-IV signal in (**B**)) or in cells expressing

*Figure 4 continued on next page*

*Figure 4 continued*

UAS::*cora*-RNAi under *pnr*-Gal4 control (Nrx-IV reduced signal in (**C**)). Yellow dashed square shows (**C'** and **C"**) magnification of cells with partial or no colocalisation between Shrb::GFP and FK2 as well as aggregates of FK2 surrounded by Shrb::GFP staining in a planar view (**C'**) and transversal view (**C"**). White square shows the magnification of cells without Nrx-IV aggregates (**C'''**). (**D–D"**) Localisation of Shrub::GFP+anti-GFP antibody and FK2 in cells expressing UAS::*Nrx-IV*-RNAi under *pnr*-Gal4 control (Nrx-IV signal disappearance in (**D**)). Yellow dashed square shows (**D'** and **D"**) magnification of cells with partial or no colocalisation between Shrb::GFP and FK2 in a planar view (**D'**) and transversal view (**D"**). (**E**) Localisation of FK2 in both wild-type and *aka*$^{L200}$ cells, separated by the dashed yellow line. Clones of wild-type and *aka*$^{L200}$ cells identified by nls::RFP marking (magenta). The scale bar represents 5 µm (**A**, **B**, **C**, **D**, and **E**) and 3 µm in (**C"** and **D"**). Dashed white lines highlight apical and basal limits of the *notum*.

junctional Crb, we observed both an enrichment at the plasma membrane associated with small punctate structures at or adjacent to the junctions using Crb::GFP in *aka*$^{L200}$ cells (**Figure 5A–A'**) and an enrichment of punctate structures at or adjacent to the plasma membrane when using the anti-Crb (**Figure 5B–B'**). In *nrv2*$^{k13315}$ cells, although junctional Crb::GFP signal was not significantly different than in control cells, the anti-Crb signal showed differences and displayed again a less well-defined pattern at the junction compared to Crb::GFP (**Figure 5—figure supplement 2A–C**). While the reason for the difference in plasma membrane and/or cortical labelling appearance between the GFP probe and antibody remains unknown at present, these observations could indicate that Crumbs is closely juxtaposed to the plasma membrane rather than residing at the plasma membrane. Interestingly and in striking contrast to Shrub depletion, we did not observe Crb and Kune basal aggregates in *aka*$^{L200}$ and *nrv2*$^{k13315}$ conditions. Hence, if both Shrub and bSJ/tSJ defects lead to Crumb altered signals, Shrub depletion is responsible for Crb being enriched in enlarged intracellular compartments whereas loss of Aka or Nrv2 triggers Crb enrichment at the apical level of the cell. Thus, as proposed above for Nrx-IV, these data further suggest a hijacking of Shrub activity towards recycling components upon alteration of SJ integrity. The elevated apical levels of Crb upon depletion of SJ component is proposed to be causal to apical enrichment of the Crumbs effector Karst (**Figure 2—figure supplement 1D–D'**; *Médina et al., 2002*). Therefore, we decided to remove one copy of Crb in the *aka*$^{L200}$ context to observe if we were able to rescue the AJ phenotype. Although we observed a rescue of the cell area phenotype (**Figure 5E**), removal of one copy of Crb was not sufficient to restore E-cad::GFP level to the control situation (**Figure 5D–D'**, 1.7-fold enrichment for bicellular junctions, 1.8-fold enrichment for TCJs).

Loss of Aka led to elevated Crb, E-cad, p-Myo-II, and Vinc::GFP signals at AJ level. In addition, Vinc-GFP staining also increased basally, with Vinc-GFP-positive structures appearing at the basolateral domain (**Figure 2—figure supplement 1A'—B**). Vinc is recruited both at AJ and in focal adhesion (FA) contact (*Kale et al., 2018*; *le Duc et al., 2010*; *Riveline et al., 2001*) and α5- and β1-integrins are regulated via the ESCRT pathway in vertebrates (*Lobert and Stenmark, 2012*). In pupal *notum*, depletion of Shrub led to accumulation of Myospheroid (Mys), the β-subunit of *Drosophila* integrin dimer, in compartments that partially colocalised with Kune (**Figure 6—figure supplement 1A–A""**), presumably enlarged endosomes, indicating that in invertebrate also, β-integrin levels rely on ESCRT-III function. In line with the hypothesis of the hijacking of Shrub activity upon depletion of SJ components, increased levels of integrin were predicted to recycle back to the plasma membrane.

Indeed, we found that Mys levels were elevated in *aka*$^{L200}$ clones, and that Mys localised in basal clusters along with F-actin (**Figure 6A–B**). Mys also colocalised with Vinc-GFP in *aka*$^{L200}$ cells, indicating an assembly of FA contacts in *aka*$^{L200}$ mutant cells (**Figure 6C–D**). Could these FA contact exert more pulling forces in *aka*$^{L200}$ cells and hence, mutant cells react by increasing their amount of apical E-cad, perhaps to sustain cell adhesion and prevent cell extrusion? To investigate this possibility, we knocked down Mys in *aka*$^{L200}$ cells. When compared to *aka*$^{L200}$ cells (**Figure 1B–B'**), depletion of Mys in *aka*$^{L200}$ cells almost abolished the E-cad enrichment at bAJs and at tAJs (**Figure 6E–F**). The cell area was also no longer significantly different than from WT (**Figure 6G**). Thus, concomitant loss of tSJ and FA contact in mature epithelium is not sufficient to induce cell extrusion. We propose that alteration of SJ integrity in pupal *notum* redirects Shrub activity to promote recycling of the junctional components Crumbs and Mys that collectively contribute to support the maintenance of epithelial integrity.

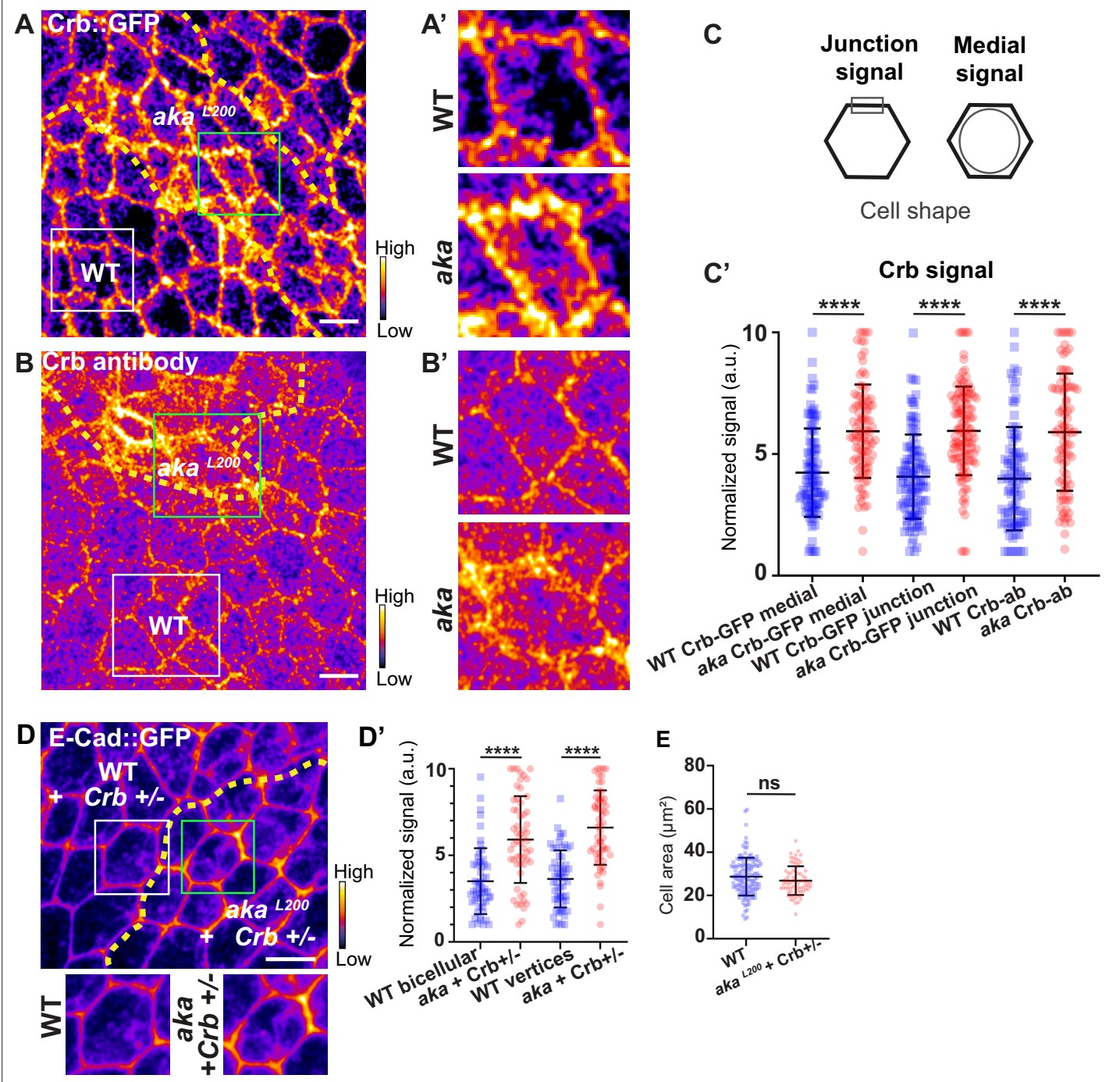

**Figure 5.** Loss of Anakonda leads to higher level of Crumbs at both junctional and medial part of the cell. (**A–B'**) show *nota*, expressing Crb::GFP (**A** and **A'**, fire colour) or stained for Crb (anti-Crb; **B** and **B'**, fire colour), between 16 hr and 19 hr after puparium formation (APF), after heat-shock to induce clone of wild-type and mutant cells for Aka. (**A–A'**) Localisation of Crb::GFP in both wild-type and *aka^{L200}* cells, separated by the dashed yellow line. (**B–B'**) Localisation of anti-Crb in both wild-type and *aka^{L200}* cells, separated by the dashed yellow line. (**C**) Scheme representing junctional and medial population of Crb staining. (**C'**) Plot of the standardised Crb::GFP signal at the medial and junctional part of the cell or anti-Crb only at the medial part, in wild-type (blue squares) and *aka^{L200}* cells (red circles) (n=100 and 96 cellular medial networks with Crb::GFP, n=110 and 119 junctions with Crb::GFP and n=90 and 88 cellular medial networks with anti-Crb for wild-type and *aka^{L200}* respectively, n=5 pupae for each condition). (**D–D'**) Localisation of E-cad::GFP (**D**, fire colour) in wild-type and *aka^{L200}* cells lacking one copy of Crb (Crb+/-). Wild-type and *aka^{L200}* cells are separated by the dashed yellow lines in (**D**). (**D'**) Plot of the standardised E-cad::GFP signal at bicellular junctions and vertices in wild-type (blue squares) and *aka^{L200}* (red circles) cells lacking one copy of Crb (n=55 and 57 bicellular junctions and n=59 and 58 vertices for wild-type and *aka^{L200}* cells respectively; n=4 pupae for each condition). (**E**) Quantification of the cell area (in μm²) of WT (wild-type) (blue squares, n=136 cells, n=4 pupae) and *aka^{L200}* cells lacking one copy of Crb

*Figure 5 continued on next page*

*Figure 5 continued*

(red circles, n=75 cells, n=4 pupae). Bars show mean ± SD, ****p<0.0001, Mann–Whitney test. A calibration bar shows LUT for grey value range. The scale bars represent 5 μm. White squares represent close-up of WT and green squares of *aka^L200* situations for panels **A**, **B**, and **D**.

The online version of this article includes the following figure supplement(s) for figure 5:

**Figure supplement 1.** Loss of function of ESCRT III protein Shrub in *notum* cells leads to Crumbs and septate junction (SJ) protein Kune-Kune abnormal localisation.

**Figure supplement 2.** Loss of Nervana 2 leads to higher level of Crumbs at adherens junction level.

## Discussion

In this study, we examined how epithelial cells can cope with and are able to remain within the tissue upon loss of SJ integrity. We report that loss of bSJs and tSJs by altering SJ integrity triggers an ESCRT-dependent response to favour bSJ transmembrane proteins recycling instead of promoting lysosomal degradation. By reducing the ESCRT-dependent degradative pathway, the cellular levels of ESCRT cargoes, including Crb and Mys, become elevated. Firstly, we propose that increased levels of Crb induce elevated Crb activity which may, at least in part, be responsible for the enhancement of apical actomyosin contractility/cellular mechanics. Secondly, FA contact points, containing Vinc and Mys, are assembled. We propose a model whereby increased Crb activity and FA contact formation may compensate for bSJ contact alteration, by reinforcing adhesion, ensuring mechanical barrier integrity (*Figure 7*).

How could SJ defects be detected?

In the pupal *notum*, the loss of tSJs leads to a loss of bSJ signal at the vertex (*Esmangart de Bournonville and Le Borgne, 2020*), weakening the three-cell contact as suggested by the holes observed by TEM [this study], presumably preventing the cells from fulfilling their paracellular diffusion barrier function. Because the observed phenotypes on E-cad, Crb, Integrin, Shrb, FK2 are cell-autonomous (only mutant cells are affected), we do not favour a model according which SJ alteration would cause indirect hormonal and/or gene expression defects at the organismal level. However, as the phenotypical consequences of SJ alteration are analysed 2–3 days after the induction of clones or gene silencing, to unambiguously demonstrate that the observed effects are a direct consequence of SJ alteration or not awaits further investigation using for instance acute methods of gene product depletion.

Keeping in mind this potential limitation, we propose, based on our previous study (*Esmangart de Bournonville and Le Borgne, 2020*) and the work of *Babatz et al., 2018*, and *Fox and Andrew, 2015*, in which SJ defects have been shown to trigger large membrane deformations, that mutant epithelial cells are capable of detecting SJ defects. Our work shows that a part of the SJ complex defects involves the ESCRT machinery. This machinery exhibits two main functions in endosomal sorting. Firstly, at the outer surface of nascent MVBs, ESCRT machinery is involved in the targeting of ubiquitinylated proteins into intraluminal vesicles, which contain the cargoes destined for lysosomal degradation. Secondly, ESCRT machinery regulates retromer-dependent recycling of bSJ components. The accumulation of the FK2 epitopes observed in this study indicates that the primary function of Shrub is attenuated upon alteration of SJ integrity, and we propose that it is in favour of the recycling function. The increased recycling of bSJ components occasioned by the loss of tSJs would thus be responsible for the large membrane deformations containing an excess of bSJ components, demonstrating a feedback between bSJs and tSJs. In contrast, the loss of bSJ components, such as Nrv2, Cora, or Nrx-IV, leads to an overall reduction in the bSJ components Cora, Nrx-IV, ATP-α, and Kune-Kune at the plasma membrane. This is explained by the fact that upon loss of a core SJ component, bSJs are not assembled into stable structures, as shown by fluorescence recovery after photobleaching analysis (*Daniel et al., 2018*; *Oshima and Fehon, 2011*). Hence, we propose that in this situation, SJ components are more recycled. We cannot exclude the possibility that the components of the SJ are partly degraded, but in this condition, the degradation would be independent of ESCRT-III.

How can SJ alteration modify Shrub function and impact intracellular trafficking? Is it due to the sensing of defects in the paracellular diffusion barrier or in cell adhesive properties, or a combination of both? It is interesting to note that numerous SJ components are GPI-anchored proteins and that, for example, *wunen-1* and *wunen-2* encode lipid phosphate phosphatase (*Ile et al., 2012*), raising the question of whether the lipid composition of the lateral plasma membrane can be altered by the loss

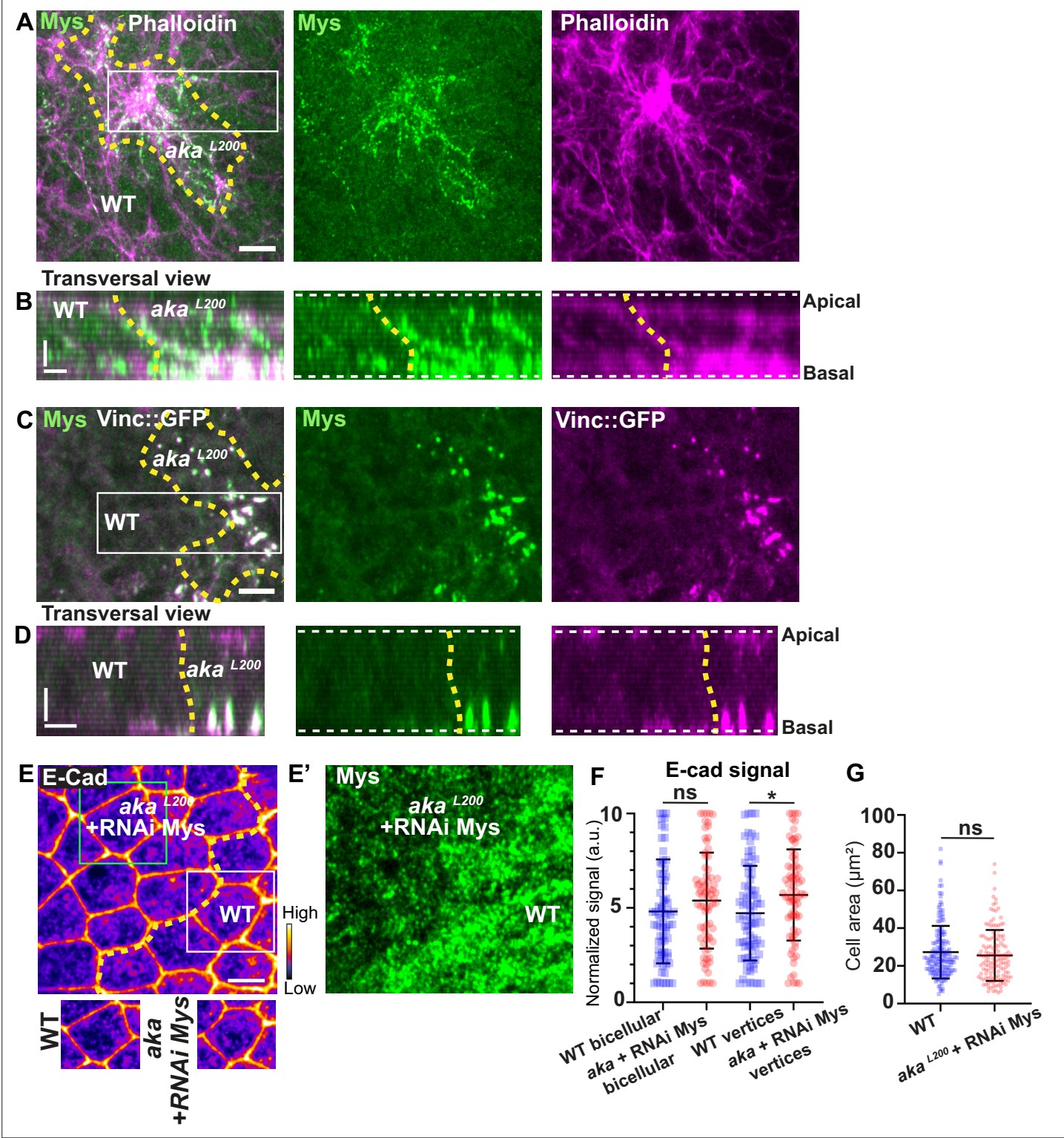

**Figure 6.** Loss of Anakonda triggers formation of focal adhesions contact. (**A–D**) show *nota* stained with Mys (**A–B**, green) and F-actin (**A–B**, Phalloidin, magenta) or expressing Vinc::GFP (**C–D**, magenta) and stained with Mys (**C–D**, green) between 16 hr and 19 hr after puparium formation (APF), after heat-shock to induce clone of wild-type and mutant cells for Aka. (**A**) Localisation of Mys (green) and F-actin (magenta) in both wild-type and *aka^L200* cells in a planar view at the basal level, separated by the dashed yellow line. (**B**) Transversal view of images depicted in **A**. (**C**) Localisation of Mys (green) and Vinc::GFP (magenta) in both wild-type and *aka^L200* cells in a planar view at the basal level, separated by the dashed yellow line. (**D**) Transversal view of images depicted in C (n>5 pupae for each condition). (**E–E'**) Localisation of E-cad (anti-E-cad; E, fire colour) and Mys stained with Mys antibody (E',

*Figure 6 continued on next page*

*Figure 6 continued*

green colour) in wild-type and *aka^L200^* cells in which Mys is knocked down (RNAi-Mys). Wild-type and *aka^L200^* cells are separated by the dashed yellow lines in (**E**). (**F**) Plot of the standardised E-cad signal at bicellular junctions and vertices in wild-type (blue squares) and *aka^L200^*+Mys knock-down cells (red circles) (n=76 and 76 bicellular junctions and n=81 and 76 vertices for wild-type and *aka^L200^* cells respectively; n>5 pupae for each condition). (**G**) Quantification of the cell area (in μm²) of WT (wild-type) (blue squares, n=171 cells, n>5 pupae) and *aka^L200^*+Mys knock-down cells (red circles, n=139 cells, n>5 pupae). Bars show mean ± SD, *p<0.05, Mann–Whitney test. A calibration bar shows LUT for grey value range. The scale bars represent 5 μm in **A** and **C** and **E** and 3 μm in **B** and **D**. White square represents close-up of WT and green square of *aka^L200^* situations for panel **E**. Dashed white lines in (**D**) highlight apical and basal limits of the *notum*.

The online version of this article includes the following figure supplement(s) for figure 6:

**Figure supplement 1.** Loss of function of Shrub in *notum* cells leads to Myospheroid and Kune-Kune abnormal localisation; related to *Figure 6*.

of SJ components. In vertebrates, the integrity of the blood–brain barrier (BBB) is regulated by the major facilitator superfamily domain containing 2a (Mfsd2a) (*Nguyen et al., 2014*). Mfsd2a is a central nervous system (CNS) endothelial-cell-specific lipid transporter that delivers the omega3-fatty acid docosahexaenoic acid into the brain via transcytosis. Lipids transported by Mfsd2a create a unique lipid composition in CNS endothelial cells that specifically inhibits caveolae-mediated transcytosis to maintain BBB integrity (*Andreone et al., 2017*). By analogy to Mfsd2a, in *Drosophila* pupal *notum*, changes in the lipid transported or in the lipid content of the plasma membrane could be sensed upon alteration of SJ integrity and modify intracellular trafficking, i.e., ESCRT-dependent recycling of SJ components. Changes in lateral plasma membrane lipid composition upon SJ alteration could also impact the lipid composition of endosomal compartments that, in turn, could participate in modulating recycling versus the degradative function of ESCRT (*Booth et al., 2021*; *Boura et al., 2012*; *Record et al., 2018*).

## Cause and consequences of SJ alteration on cell mechanics and adhesion

We do not favour a model in which the effects on cell mechanics and adhesive properties resulting from altered SJ integrity can be caused by a defect in cell polarity. Indeed, loss of cell polarity regulators Scrib/Dlg causes delocalisation of E-cad/Arm and of Crb to the basolateral part of the cell (*Bilder et al., 2000*), phenotypes we never observed upon loss of SJ components including Aka, M6, Gli, Nrv2, and Cora. Here, in both bSJ and tSJ mutant cells, Crb is enriched at the apical pole of the cells. This might be the result of an overall increased Crb transcription levels. However, in the event of a transcriptional response, this would be a global effect on gene expression since E-cad, Crb, Mys, and bSJ components levels are also increased. Hence, although we cannot firmly exclude a global effect at transcriptional level, we favour the hypothesis of increased recycling.

As Crb is a known binding partner of the β-Heavy Spectrin Karst (*Médina et al., 2002*), Crb defects are proposed to cause the enrichment of Karst in the bSJ/tSJ mutant cells. Furthermore, the enrichment of Myo-II::GFP, and especially p-Myo-II, might be due to the upregulation of the activator Rho-kinase (Rok), another known partner of Crb (*Sidor et al., 2020*). Interestingly, the *Drosophila* tSJ protein M6 has been recently reported to act as an interplay partner of Ajuba (*Ikawa et al., 2023*) and loss of M6 is associated with elevated signal of Ajuba at vertices in pupal wing epithelium. The fact that we similarly observed elevated signal of Ajuba upon loss of Aka in the pupal *notum* reinforces the idea of AJ remodelling by mechanistic links between tSJ and AJ/actomyosin cytoskeleton components.

The formation of shorter cell–cell interfaces during cytokinesis in *aka* mutant cells could argue for changes in tensile forces. However, these short interfaces could also result from high contractile forces within the cytokinetic ring and reduced resistance from neighbours. It can also be the consequence of delayed E-cad disengagement due to higher levels of E-cad, as reported in *Drosophila* embryos (*Guillot and Lecuit, 2013*), rather than an overall change in tissue tensile forces. In fact, we did not observe differences in the recoil velocity of *aka* or *nrv2* mutant cells upon junction nanoablation. A plausible explanation seems related to the fact that all mutant cells have their level of medial and junctional Myo-II and linked AJs increased. Therefore, the pulling forces might be at equilibrium as in WT condition, and might be equal on both sides of the junctions. An argument in favour of similar tension in both WT/heterozygous and mutant cells is that clones of mutant cells are compact and do

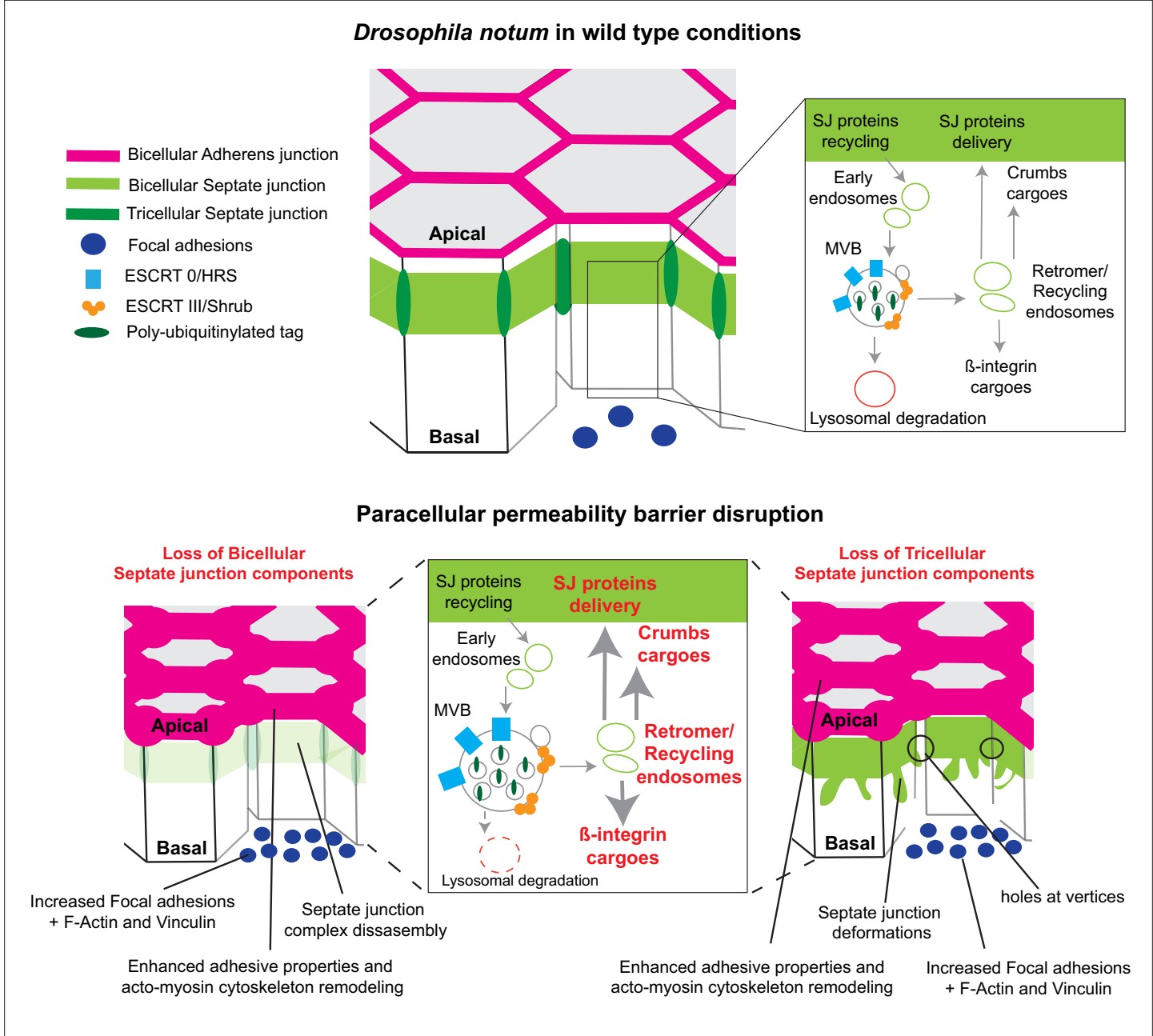

**Figure 7.** Model summarising the effects of the disruption of septate junction (SJ) integrity in pupal notum. In wild-type conditions, bicellular SJ (bSJ) proteins, β-integrin, and Crumbs are recycled to the membrane, thanks to the endosomal–retromer complex. When the paracellular permeability function is compromised due to the loss of bSJ or tricellular SJ (tSJ) components, cells favour recycling over degradation, leading to increased levels of β-integrin and Crumbs at the cell membrane. The accumulation of β-integrin and Crumbs leads to a strengthening of the adhesive structure as shown by increased quantity of adherens junction (AJ) proteins but also by the appearance of focal adhesion contacts. We propose that the cell compensates the lack of bSJ contacts by increasing its adhesive properties.

not disperse among WT cells (or vice versa). One could expect mixing of cells upon differential tension at boundaries, as highlighted in *Levayer et al., 2015*.

In any case, despite the increased amount of E-cad and Myo-II, *aka* mutant cells do not undergo apical constriction, basal cell extrusion nor induce a fold in the tissue. Thus, the changes observed argue in favour of reinforcing adhesion to prevent cell extrusion. Another argument in favour of a reinforcement of adhesive properties upon SJ alterations is the assembly of Vinc and Mys FAs laterally and basally. Although FA contacts, restricted to the basal site, are present at the location of

attachment sites of flight muscles, rather late in pupal development (*Lemke and Schnorrer, 2017*), it is notable that FAs are being detected in *aka* clones as early as 15–16 hr after puparium formation (APF). Although Mys and Vinc are expressed in control epithelial cells, they do not assemble into detectable FAs. Although, as mentioned above, we cannot exclude the possibility that this is due to transcriptional upregulation of Mys, we favour the hypothesis that reduced degradation of Mys by the ESCRT machinery contributes to FA assembly. We propose that such contacts contribute to the maintenance of epithelial cells within the epithelium layer, hence contributing to mutant cells' preservation of epithelial mechanical integrity upon SJ disturbance.

## Conservation of the process?

Does a similar detection mechanism exist in vertebrates upon alteration of TJs? Of note, in *Xenopus* embryos, leaks at TJs occurring as cell boundaries elongate are detected and induce transient and local activation of Rho, named 'Rho flares' (*Stephenson et al., 2019*). This leads to localised contraction of the cell boundary to restore the local concentration of TJ proteins (*Stephenson et al., 2019*). During the course of our study, Rho-flare formation was shown to be mechanically triggered by mechanosensitive calcium-channel-dependent calcium flashes in TJ remodelling (*Varadarajan et al., 2022*). This mechanism permits epithelium to repair small TJ leaks induced by mechanical stimuli. Whether a similar mechanosensitive-dependent repair mechanism is at play in *Drosophila* requires further investigation. If so, how does this compare with the mechanism described in our study?

Another recent study revealed that serine proteinases are used to cleave the TJ complex form by proteins EpCAM and Claudin-7 upon TJ damages, releasing Claudin-7 and ensuring TJ rapid repair (*Higashi et al., 2023*). Conversely, it remains to be determined whether such mechanisms described for small leaks apply to larger alterations of the TJ belt, as we report here in flies, and involve AJ and FA reinforcement of adhesive properties.

Due to their importance in ensuring epithelia homeostasis, deciphering between direct and indirect consequences of TJ alterations remains a key question to explore in the future.

## Experimental model

### *Drosophila* genotypes

***Figure 1*** (A–A') UAS-Aka-RNAi-TRiP; *pnr*-Gal4 obtained by crossing UAS-Aka-RNAi-TRiP with *pnr*-Gal4/TM6, *Tb1*. (B–B') *hs*-FLP; *aka*$^{L200}$, FRT40A/*ubi*-RFP nls, FRT40A, E-cad::GFP; obtained by crossing *hs*-FLP; *aka*$^{L200}$, FRT40A/CyO with *hs*-FLP; *ubi*-RFP nls, FRT40A, ECad::GFP/CyO. (C–C') *hs*-FLP; Myo-II::GFP; *ubi*-RFP nls, FRT40A/CyO obtained by crossing Myo-II::GFP; *ubi*-RFP nls, FRT40A/CyO; with *hs*-FLP; *aka*$^{L200}$, FRT40A/CyO. (D–D') *hs*-FLP; *aka*$^{L200}$, FRT40A/*ubi* -RFP nls, FRT40A obtained by crossing *hs*-FLP; *aka*$^{L200}$, FRT40A/CyO with *hs*-FLP; *ubi*-RFP nls, FRT40A/(CyO). (E–E') *hs*-FLP; *nrv2*$^{k13315}$, FRT40A/*ubi* -RFP nls, FRT40A, E-cad::GFP; obtained by crossing *hs*-FLP; *nrv2*$^{k13315}$, FRT40A/CyO with *hs*-FLP; *ubi*-RFP nls, FRT40A, E-cad::GFP/CyO (F-F') *hs*-FLP/Myo-II::GFP; *nrv2*$^{k13315}$, FRT40A/*ubi*-RFP nls, FRT40A obtained by crossing Myo-II::GFP; *ubi*-RFP nls, FRT40A/CyO; with *hs*-FLP; *nrv2*$^{k13315}$,-FRT40A/CyO (G-G') *hs*-FLP; *cold*$^{f05607}$, FRT40A/*ubi*-RFP nls, FRT40A, E-cad::GFP; obtained by crossing *hs*-FLP; *cold*$^{f05607}$, FRT40A/CyO with *hs*-FLP; *ubi*-RFP nls, FRT40A, ECad::GFP/CyO.

***Figure 2*** (A–B) *hs*-FLP; Myo-II::GFP; *ubi*-RFP nls, FRT40A/CyO obtained by crossing Myo-II::GFP; *ubi*-RFP nls, FRT40A/CyO; with *hs*-FLP; *aka*$^{L200}$, FRT40A/CyO (E-F) *hs*-FLP; *aka*$^{L200}$, FRT40A/*ubi*-RFP nls, FRT40A, E-cad::GFP; obtained by crossing *hs*-FLP; *aka*$^{L200}$, FRT40A/CyO with *hs*-FLP; *ubi*-RFP nls, FRT40A, ECad::GFP/CyO.

***Figure 3*** (A–D') Shrub::GFP; UAS-cora-RNAi/UAS-KAEDE, *pnr*-Gal4 obtained by crossing UAS-cora-RNAi with; Shrub::GFP; UAS-KAEDE, *pnr*-Gal4/SM5-TM6, *Tb1*.

***Figure 4*** (A) *sca*-Gal4/UAS-Shrub-RNAi-TRiP obtained by crossing;; *sca*-Gal4 with;; UAS-Shrub-RNAi-TRiP. (B–C''') Shrub::GFP; UAS-cora-RNAi /*pnr*-Gal4 obtained by crossing UAS-cora-RNAi with; Shrub::GFP; *pnr*-Gal4/SM5-TM6, *Tb1*. (D–D'') Shrub::GFP; UAS-Nrx-IV-RNAi/*pnr*-Gal4 obtained by crossing UAS-Nrx-IV-RNAi with Shrub::GFP; *pnr*-Gal4/TM6, *Tb1*. (E) *hs*-FLP; *aka*$^{L200}$, FRT40A/*ubi*-RFP nls, FRT40A obtained by crossing *hs*-FLP; *aka*$^{L200}$, FRT40A/CyO with *hs*-FLP; *ubi*-RFP nls, FRT40A/(CyO).

***Figure 5*** (A–A') *hs*-FLP; *aka*$^{L200}$, FRT40A/*ubi*-RFP nls, FRT40A; Crb::GFP/+obtained by crossing *ubi*-RFP nls, FRT40A/CyO; Crb::GFP/TM6, *Tb1* with *hs*-FLP; *aka*$^{L200}$, FRT40A/CyO. (B–B') *hs*-FLP; *aka*$^{L200}$, FRT40A/*ubi* -RFP nls, FRT40A obtained by crossing *hs*-FLP; *aka*$^{L200}$, FRT40A/CyO with *hs*-FLP;

*ubi*-RFP nls, FRT40A/(CyO). (D–D') *hs*-FLP; *aka^L200^*, FRT40A/*ubi*-RFP nls, FRT40A, E-cad::GFP; FRT82B, Crb^11a22^/+obtained by crossing; *aka^L200^*, FRT40A/CyO; FRT82B, Crb^11a22^/TM6, *Tb1* with *hs*-FLP; *ubi*-RFP nls, FRT40A, ECad::GFP/CyO.

*Figure 6* (A–D) *hs*-FLP; *aka^L200^*, FRT40A/*ubi*-RFP nls, FRT40A obtained by crossing *hs*-FLP; *aka^L200^*, FRT40A/CyO with *hs*-FLP; *ubi*-RFP nls, FRT40A/(CyO). (E–E') *hs*-FLP, UAS-GFP; *aka^L200^*, FRT40A/*tub*-GAL80, FRT40A; UAS-Mys-RNAi-TRiP/*tub*-GAL4 obtained by crossing *aka^L200^*, FRT40A; UAS-Mys-RNAi-TRiP /SM5-TM6b, *Tb1* with *hs*-FLP, UAS-GFP, tub-GAL80, FRT40A; tub-GAL4/TM6C, *Sb₁, Tb1*.

*Figure 1—figure supplement 1* (A–A") *hs*-FLP; *aka^L200^*, FRT40A/*ubi*-RFP nls, FRT40A, E-cad::GFP; obtained by crossing *hs*-FLP; *aka^L200^*, FRT40A/CyO with *hs*-FLP; *ubi*-RFP nls, FRT40A, ECad::GFP/CyO. (B) *hs*-FLP; *Gli^dv3^*, FRT40A/*ubi*-RFP nls, FRT40A; obtained by crossing *hs*-FLP; *Gli^dv3^*, FRT40A/CyO with *hs*-FLP; *ubi*-RFP nls, FRT40A/CyO. (C–C') *hs*-FLP; *aka^L200^*, FRT40A/*ubi*-RFP nls, FRT40A, E-cad::GFP; obtained by crossing *hs*-FLP; *aka^L200^*, FRT40A/CyO with *hs*-FLP; *ubi*-RFP nls, FRT40A, ECad::GFP/CyO.

*Figure 2—figure supplement 1* (A–B) *hs*-FLP; *aka^L200^*, FRT40A/*ubi*-RFP nls, FRT40A; Vinc::GFP/+obtained by crossing *ubi*-RFP nls, FRT40A/CyO; Vinc::GFP/TM6, *Tb1* with *hs*-FLP; *aka^L200^*, FRT40A/CyO. (D–D') *hs*-FLP; *aka^L200^*, FRT40A/*ubi*-RFP nls, FRT40A; Karst::YFP/+obtained by crossing; *ubi*-RFP nls, FRT40A / CyO; Karst::YFP/TM6, *Tb1* with *hs*-FLP; *aka^L200^*, FRT40A/CyO. (E–E') *hs*-FLP; *aka^L200^*, FRT40A/*ubi*-RFP nls, FRT40A; Jub::GFP /+obtained by crossing; *ubi*-RFP nls, FRT40A/CyO; Jub::GFP/TM2 with *hs*-FLP; *aka^L200^*, FRT40A/CyO.

*Figure 3—figure supplement 1* (A–A') *hs*-FLP; *aka^L200^*, FRT40A/*ubi*-RFP nls, FRT40A obtained by crossing *hs*-FLP; *aka^L200^*, FRT40A/CyO with *hs*-FLP; *ubi*-RFP nls, FRT40A/(CyO). (C–C') *hs*-FLP; *nrv2^k13315^*, FRT40A/*ubi*-RFP nls, FRT40A; obtained by crossing *hs*-FLP; *nrv2^k13315^*, FRT40A/CyO with *hs*-FLP; *ubi*-RFP nls, FRT40A/CyO.

*Figure 5—figure supplement 1* (A–A"") *sca*-Gal4/UAS-Shrub-RNAi-TRiP obtained by crossing;; *sca*-Gal4 with;; UAS-Shrub-RNAi-TRiP.

*Figure 5—figure supplement 2* (A–A') *hs*-FLP; *nrv2^k13315^*, FRT40A/*ubi*-RFP nls, FRT40A; Crb::GFP/+obtained by crossing; *ubi*-RFP nls, FRT40A/CyO; Crb::GFP/TM6, *Tb1* with *hs*-FLP; *nrv2^k13315^*, FRT40A/CyO. (B–B') *hs*-FLP; *nrv2^k13315^*, FRT40A/*ubi*-RFP nls, FRT40A; obtained by crossing *hs*-FLP; *nrv2k13315*, FRT40A/CyO with *hs*-FLP; *ubi*-RFP nls, FRT40A/CyO.

*Figure 6—figure supplement 1* (A–A"") *sca*-Gal4/UAS-Shrub-RNAi-TRiP obtained by crossing;; *sca*-Gal4 with;; UAS-Shrub-RNAi-TRiP.

## Materials and methods

### Key resources table

| Reagent type (species) or resource | Designation | Source or reference | Identifiers | Additional information |
|---|---|---|---|---|
| Genetic reagent (*Drosophila melanogaster*) | Myo-II::GFP^crispr^ | *Esmangart de Bournonville and Le Borgne, 2020* | N/A | |
| Genetic reagent (*D. melanogaster*) | *hs*-FLP; *aka^L200^*, FRT40A / CyO | *Esmangart de Bournonville and Le Borgne, 2020* | N/A | |
| Genetic reagent (*D. melanogaster*) | ; E-cad::GFP; | *Huang et al., 2009* | N/A | |
| Genetic reagent (*D. melanogaster*) | ; *nrv2^k13315^*, FRT40A / CyO | *Chen et al., 2005* | DGRC 114351 | |
| Genetic reagent (*D. melanogaster*) | ; *cold^f05607^*, FRT40A / CyO | Kyoto Stock Center | Stock: 114 662 | |
| Genetic reagent (*D. melanogaster*) | ; *Gli^dv3^*, FRT40A / CyO | *Schulte et al., 2003* | Gift from Vanessa Auld | |
| Genetic reagent (*D. melanogaster*) | ;; FRT 82B, Crb ^11A22^ | *Tepaß and Knust, 1990* | Gift from Ulrich Tepass | |
| Genetic reagent (*D. melanogaster*) | *hs*-FLP, UAS-GFP, y[1] w[*]; tub-GAL80 FRT40A; tub-GAL4/TM6C, *Sb₁, Tb₁* | *Lee and Luo, 2001* | BDSC Stock: 5192 | |

*Continued on next page*

*Continued*

| Reagent type (species) or resource | Designation | Source or reference | Identifiers | Additional information |
|---|---|---|---|---|
| Genetic reagent (*D. melanogaster*) | ; Shrub::GFP/CyO; | N/A | Gift from Juliette Mathieu | |
| Genetic reagent (*D. melanogaster*) | w;; Jub::GFP/TM2 | *Rauskolb et al., 2014* | BDSC Stock: 56806 | |
| Genetic reagent (*D. melanogaster*) | w;; UAS-KAEDE | BDSC | BDSC Stock: 26161 | |
| Genetic reagent (*D. melanogaster*) | ;; Crb::GFP | *Huang et al., 2009* | Crb::GFP-A GE24 | |
| Genetic reagent (*D. melanogaster*) | ;; Vinc::GFP | *Kale et al., 2018* | Gift from Thomas Lecuit | |
| Genetic reagent (*D. melanogaster*) | w[1118];; Karst::YFP | Kyoto Stock Center | Stock: 115 518 | |
| Genetic reagent (*D. melanogaster*) | *hs*-FLP; *ubi*-RFP nls, FRT40A / (CyO) | *Claret et al., 2014* | Gift from Antoine Guichet | |
| Genetic reagent (*D. melanogaster*) | ; UAS-Aka-RNAi-TRiP; | *Perkins et al., 2015* | BDSC Stock: 67014 | |
| Genetic reagent (*D. melanogaster*) | ; UAS-Nrx-IV-RNAi; | VDRC | Stock: 108 128 | |
| Genetic reagent (*D. melanogaster*) | ;; UAS-Cora-RNAi-TRiP | *Perkins et al., 2015* | BDSC Stock: 9788 | |
| Genetic reagent (*D. melanogaster*) | ;; UAS-Shrub-RNAi-TRiP | *Perkins et al., 2015* | BDSC Stock: 38305 | |
| Genetic reagent (*D. melanogaster*) | ;; UAS-Mys-RNAi-TRiP/TM3, Sb | *Perkins et al., 2015* | BDSC Stock: 27735 | |
| Genetic reagent (*D. melanogaster*) | ;; *pnr*-Gal4/TM6, *Tb[1]* | *Calleja et al., 1996* | N/A | |
| Genetic reagent (*D. melanogaster*) | ;; *sca*-Gal4 | *Mlodzik et al., 1990* | N/A | |
| Antibody | Anti-Coracle (Mouse, monoclonal) | DSHB | C615.16, RRID:AB_1161644 | (1:200) |
| Antibody | Anti-E-cad (Rat, monoclonal) | DSHB | DCAD2; AB_528120 | (1:500) |
| Antibody | Anti-Kune (Rabbit, polyclonal) | *Nelson et al., 2010* | Gift from Mikio Furuse | (1:1000) |
| Antibody | Anti-HRS (Mouse, monoclonal) | DSHB | 27-4 | (1:100) |
| Antibody | Anti-Nrx-IV (Rabbit, polyclonal) | *Stork et al., 2009* | Gift from Christian Klämbt | (1:1000) |
| Antibody | Anti-GFP (Goat, polyclonal) | Abcam | Cat#ab5450 | (1:1000) |
| Antibody | Anti-FK2 (Mouse, monoclonal) | Sigma-Aldrich | Cat#04-263 | (1:1000) |
| Antibody | Anti-Crb (Rat, polyclonal) | *Richard et al., 2006* | Gift from Elisabeth Knust | (1:1000) |
| Antibody | Anti-Mys (Rabbit, monoclonal) | DSHB | CF.6G1, RRID:AB_528310 | (1:200) |
| Antibody | Anti phospho-Myo-II (Mouse, monoclonal) | Cell Signalling | Cat#mab 3675 | (1:1000) |
| Antibody | Cy2-, Cy3-, and Cy5-coupled secondary antibodies | The Jackson Laboratory | N/A | (1:300) |
| Antibody | Alexa Fluor 647 Phalloidin | Thermo Fisher Scientific | Cat#A22287 | (1:1000) |
| Chemical compound, drug | Paraformaldehyde | EMS | 19340-72 | |
| Chemical compound, drug | Triton X-100 | Euromedex | 2000B | |
| Chemical compound, drug | Phosphate Buffered Saline | Lonza | BE17-515F | |

*Continued on next page*

*Continued*

| Reagent type (species) or resource | Designation | Source or reference | Identifiers | Additional information |
|---|---|---|---|---|
| Chemical compound, drug | Voltalef | VWR | 24627.188 | |
| Software, algorithm | Fiji | *Schindelin et al., 2012* | https://imagej.net/Fiji | |
| Software, algorithm | Prism 8 | GraphPad | GraphPad RRID:SCR_002798 | |
| Software, algorithm | RStudio | RStudio Team (2020). RStudio: Integrated Development for R. RStudio, PBC, Boston | http://www.rstudio.com RRID:SCR_000432 | |
| Software, algorithm | MATLAB | MATLAB and Statistics Toolbox Release 2012b | The MathWorks, Inc, Natick, MA, USA RRID:SCR_001622 | |
| Other | Confocal Microscope | Leica | LSM TCS SPE, TCS SP5 and TCS SP8 | |
| Other | Confocal Microscope | Zeiss | Confocal LSM 880 Airyscan | |

## Transmission electron microscopy sample preparation

WT and aka RNAi *Drosophila* pupal nota (16 hr APF) were dissected in 0.1 M cacodylate buffer at room temperature and immediately processed (*Kolotuev, 2014*). Briefly, the samples were fixed in 1% paraformaldehyde and 2.5% glutaraldehyde in 0.1 M cacodylate buffer for 2 hr. Then, they were stained for 1 hr in 2% (wt/vol) osmium tetroxide and 1.5% (wt/vol) K4[Fe(CN)6] in cacodylate buffer followed by 1 hr in 1% (wt/vol) tannic acid in 100 mM cacodylate buffer. Finally, they were incubated for 30 min in 2% (wt/vol) osmium tetroxide followed by 1% (wt/vol) uranyl acetate for 2 hr. After the dehydration cycles, samples were embedded in Epon–Araldite mix. To ensure precise orientation and access to the samples, a two-step flat-embedding procedure was used (*Kolotuev, 2014*). Sections were cut with an Ultracut E microtome (Reichert-Jung, Austria, now Leica Microsystems) parallel to the plane of the pupal nota epithelia. Semi-thin sections (0.7–1 μm thick) were mounted on microscope slides and stained with 1% aqueous solution of methylene blue in 1% borax. Ultrathin sections (70–80 nm thick) were collected on either standard copper grids or single-slot nickel grids coated with formvar (polyvinyl formal; Polysciences, Inc). The sections were contrasted with saturated aqueous uranyl acetate solution for 20 s, rinsed with double-distilled water, and stained in Reynolds solution (lead citrate; Sigma-Aldrich) for up to 3 min (*Reynolds, 1963*). After several rinses with deionised water and drying, the sections were examined with a JEM-2100 HT (JEOL Ltd, Japan) transmission electron microscope at 80 kV. The pupal nota of three WT and three aka RNAi specimens were examined, and at least 10 ultrathin sections of the region of interest were analysed in each specimen. At least three TCJs were examined in each set of sections.

## Immunofluorescence

Pupae aged from 16 hr 30 min to 19 hr APF were dissected using Cannas microscissors (Biotek, France) in 1× phosphate-buffered saline (1× PBS, pH 7.4) and fixed 15 min in 4% paraformaldehyde at room temperature (*Gho et al., 1996*). Following fixation, dissected nota were permeabilised using 0.1% Triton X-100 in 1× PBS (PBT), incubated with primary antibodies diluted in PBT for 2 hr at room temperature. After three washes of 5 min in PBT, nota were incubated with secondary antibodies diluted in PBT for 1 hr, followed by two washes in PBT, and one wash in PBS, prior mounting in 0.5% N-propylgallate dissolved in 90% glycerol/PBS 1× final.

## Genetics tools

37° heat shocks to induce clones of WT, heterozygous, and mutant cells were performed at L2 and L3 larval stages for an hour each time. The RNAis were driven using *pnr* and *Sca-GAL4* drivers and their expression is initiated at L3 stage. *Sca-GAL4* driver was chosen over *pnr* when the driven RNAi was lethal for the pupa. Mutant cells were analysed 2–3 days after the induction of clones or gene silencing.

## Live imaging and image analyses

Live imaging was performed on pupae aged for 16 hr 30 min APF at 25°C. Pupae were sticked on a glass slide with a double-sided tape, and the brown pupal case was removed over the head and dorsal thorax using microdissection forceps. Pillars made of four and five glass coverslips were positioned at the anterior and posterior side of the pupae, respectively. A glass coverslip covered with a thin film of Voltalef 10S oil is then placed on top of the pillars such that a meniscus is formed between the dorsal thorax of the pupae and the glass coverslip (*Gho et al., 1999*). Images were acquired with an LSM Leica SPE, SP5, or SP8 equipped with a 63× NA 1.4 objective and controlled by LAS AF software or by LSM Zeiss 880 AiryScan equipped with a 63× NA 1.4 objective and controlled by ZEN software. Confocal sections (z) were taken every 0.5 μm or 1 μm. For figures representation, images were processed with Gaussian Blur σ=1.1. All images were processed and assembled using Fiji software (*Schindelin et al., 2012*) and Adobe Illustrator.

## Nanoablation

Laser ablation was performed on live pupae aged for 16 hr to 19 hr APF using a Leica SP5 confocal microscope equipped with a 63× NA 1.4 objective or an LSM Zeiss 880 AiryScan equipped with a 63× NA 1.4 objective. Ablation was carried out on epithelial cell membranes at AJ level with a two-photon laser-type Mai-Tai HP from Spectra Physics set to 800 nm and a laser power of 2.9 W.

## Quantification and statistical analysis

### Fluorescence signal analysis

Sum slices were applied to different experiments. A circular ROI of 2 μm*2 μm was drawn to measure signal at vertices, a circular ROI of 3 μm*3 μm for the medial network and centred in the measured cells and a segmented line of 10 pixels width was used to measure signals at bicellular junctions. Using the same width or diameter, lines and circular ROI were drawn to extract background fluorescence signals and the background signal was subtracted to each quantification. After, data were normalised between 1 and 10 to allow visual representation with 10 corresponding to the highest signal in each experiment analysed and 1 the lowest. Normalisation was operated on data of cells belonging to the same *notum* in every experiment.

### Cell area quantification

Sum slices projection was applied then WT and *aka^{L200}* cells were discriminated on their presence/absence of nls::RFP signal. We excluded cells at the border of the WT/*aka^{L200}* clonal area. A mask was applied based on the E-cad::GFP or E-cad signal and area in μm² was extracted. Appropriate statistical tests were used to check for significant differences.

### Length establishment measurement

The time t=0 was set according to the frame just before the beginning of the contraction of the cell. Each frame was separated by 2 min. The maximal expected size of the junction was inferred at the beginning of the contraction with the expected localisation of the two future vertices. Then, each frame, the length was measured at the new vertices formed and standardised to the initial maximal expected size.

### Statistical tests

All information concerning the statistical details are provided in the main text and in figure legends, including the number of samples analysed for each experiment. Prism 8 software and R 4.2.1 were used to perform the analyses. No statistical tests were used to predetermine sample size. Replicates numbers were decided from experience of the techniques performed and practical considerations. No data were excluded.

Scattered plots use the following standards: thick line indicate the means and errors bars represent the standard deviations. Boxplots with connected line use the following standards: dots represent mean and the total-coloured areas show SD.

The Shapiro–Wilk normality test was used to confirm the normality of the data and the F-test to verify the equality of SD. The statistical difference of data sets was analysed using the Student's

unpaired two-tailed t test, Multiple t tests, Fisher t test, or the non-parametric Wilcoxon–Mann–Whitney test. Statistical significances were represented as follows: p-value >0.05 NS (not significant), p-value ≤0.05*; p-value ≤0.01**; and p-value ≤ 0.0001 ****.

## Acknowledgements

We thank A Guichet, C Klämbt, E Knust, T Lecuit, S Luschnig, J Mathieu, J Treisman, K Röper, and A Uv for reagents. We also thank JR Huynh and J Mathieu (CIRB, Paris) for sharing the ShrubGFP CRISPR line prior to publication and the Bloomington Stock Center, the Vienna *Drosophila* RNAi Center and the National Institute of Genetics Fly Stock Center for providing fly stocks. We also thank S Dutertre and X Pinson from the Microscopy Rennes Imaging Center-BIOSIT (France). We are grateful to A Dupont and A Jankowska for excellent technical support for the electron microscopy. The JEM 2100 HT transmission electron microscope was available at the Laboratory of Microscopy, Department of Cell Biology and Imaging, Institute of Zoology and Biomedical Research, Jagiellonian University. The monoclonal antibodies against Cora and E-cad were obtained from the Developmental Studies Hybridoma Bank, generated under the auspices of the National Institute of Child Health and Human Development, and maintained by the University of Iowa Department of Biological Sciences. This work was supported in part by a research grant (N18/DBS/000013 to MKJ) and in part by the Fondation pour la Recherche Médicale (grant number FDT202001010770 to TEdB), the Association Nationale de la Recherche programme PRC Vie, santé et bien-être ACTriCE (ANR-20-CE13-0015 to RLB), and Fondation ARC (PJA 20191209388 to RLB).

## Additional information

### Funding

| Funder | Grant reference number | Author |
|---|---|---|
| Fondation pour la Recherche Médicale | FDT202001010770 | Thomas Esmangart de Bournonville |
| Agence Nationale de la Recherche | ANR-20-CE13-0015 | Roland Le Borgne |
| Fondation ARC pour la Recherche sur le Cancer | PJA 20191209388 | Roland Le Borgne |
| The Ministry of Sciences and Higher Education for the Jagiellonian University in Krakow, Poland | N18/DBS/000013 | Mariusz K Jaglarz |

The funders had no role in study design, data collection and interpretation, or the decision to submit the work for publication.

### Author contributions

Thomas Esmangart de Bournonville, Conceptualization, Formal analysis, Funding acquisition, Investigation, Visualization, Methodology, Writing – original draft, Writing – review and editing; Mariusz K Jaglarz, Formal analysis, Funding acquisition, Investigation, Visualization; Emeline Durel, Formal analysis, Visualization, Methodology; Roland Le Borgne, Conceptualization, Formal analysis, Supervision, Funding acquisition, Validation, Investigation, Visualization, Methodology, Writing – original draft, Project administration, Writing – review and editing

### Author ORCIDs

Thomas Esmangart de Bournonville (iD) https://orcid.org/0000-0001-6012-1726
Mariusz K Jaglarz (iD) https://orcid.org/0000-0002-1606-8339
Roland Le Borgne (iD) http://orcid.org/0000-0001-6892-278X

### Decision letter and Author response

Decision letter https://doi.org/10.7554/eLife.91246.sa1
Author response https://doi.org/10.7554/eLife.91246.sa2

## Additional files

### Supplementary files
• MDAR checklist

### Data availability

All data generated or analysed during this study are included in the manuscript and the supporting data files have been made available on Dryad and includes the data set https://dx.doi.org/10.5061/dryad.dbrv15f7h. This dataset includes original stacks of confocal images from Figure 1B-D and E-G, Figure 2A, B, E and F, Figure 3A-D, Figure 4A-E, Figure 5A, A', B, B', and D, Figure 6 A-E', Figure 1 S1 A-B and C, C', Figure 2 S1 A-B, D and E, Figure 3 S1 A, A' and C, C', Figure 5 S1 A-B', Figure 5 S2 A, B', and Figure 6 S1 A-A' (including as well the confocal stacks used for quantification and statistical analyses); and detailed statistical analyses (Excel tables or Rtables) of Figure 1B'-D' and E'-G', Figure 2A', C, D and G, Figure 5C', D' and E, Figure 6F and G, Figure 1 S1 B', D and E, Figure 2 S1 C, D' and E', Figure 3 S1 B, D, Figure 5 S2 C and S6.

The following dataset was generated:

| Author(s) | Year | Dataset title | Dataset URL | Database and Identifier |
|---|---|---|---|---|
| Le Borgne R | 2024 | Data from: ESCRT-III-dependent adhesive and mechanical changes are triggered by a mechanism detecting alteration of Septate Junction integrity in Drosophila epithelial cells | https://dx.doi.org/10.5061/dryad.dbrv15f7h | Dryad Digital Repository, 10.5061/dryad.dbrv15f7h |

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
