## [Editor Report]

The authors explore an interesting question: how do epithelial tissues respond to loss of barrier function in vivo? These important results break new ground in looking at the dynamic relationships between junctional complexes. The results of this convincing paper will be of interest to a broad audience of cell and developmental biologists.

---

## [Decision Letter]

[Editors' note: this paper was reviewed by Review Commons.]

Thank you for resubmitting your work entitled "ESCRT-III-dependent adhesive and mechanical changes are triggered by a mechanism detecting alteration of Septate Junction integrity in *Drosophila* epithelial cells" for further consideration by *eLife*. Your revised article has been evaluated by Utpal Banerjee (Senior Editor) and the original reviewers.

The small number of concerns raised by the reviewers in the second round of reviews are legitimate and should be addressed with further revision. But it is important to note that fundamentally, the paper presents important work that is generally well substantiated.

*Reviewer #3 (Recommendations for the authors):*

The authors have addressed most of my comments.

The issue about the TEM images remains, as "cell membrane detachment" cannot be seen clearly on the image in Figure 1A'. The authors should rephrase their following conclusions accordingly:

Line 143: "… we report that, depletion of Aka induces weaknesses in the integrity of the tissue which results in cell membrane detachment at the vertex in the plane of the SJ with the formation of sizeable intercellular gaps within the epithelium (Figure 1A-A')."

Line 390: "… weakening the three-cells contact ultimately leading to gaps [this study)…".

Also, authors should mention that plasma membrane-lined delaminations in aka or in Gli mutant epithelia were described previously (Schulte et al. J Cell Biol 2003; Hildebrandt et al. Dev Biol 2015; Byri et al. Dev Cell 2015).

The discussion still contains rather extensive speculation, which should be marked as such, e.g., the subheading on line 387 ("A mechanism detecting bicellular septate junction defects"). Given the largely speculative nature of the arguments discussed here, the authors should consider changing the wording, e.g., to "How could septate junction defects be detected?".

It is not clear to me how the fact that SJ components include GPI-anchored proteins would support the idea that (line 428) "the lipid composition of the lateral plasma membrane is likely to be affected upon the loss of SJ components.". Please either explain or remove this point.

---

## [Author Response]

1. General Statements

We thank the three reviewers for their constructive and valuable comments. Following the criticisms, we rewrote the manuscript in order to :

Clarify the function of septate junctions (permeability barrier and more structural role linked to cell adhesion), so we do not attribute the defects to exclusively permeability barrier functionClarify the septate junction defect in relation to cell polarity defects

Add quantification of EM analysis (Figure 1)

2. Point-by-point description of the revisionsReviewer 1:The paper is completely focused on the septate junctions as a paracellular diffusion barrier. However, many of the septate junction components, including Scribble, Dlg, and Lgl, have well documented (if poorly understood) basolateral polarity functions, and considering that septate junctions contain 15 or more cell-cell adhesion proteins, they are also likely to have a adhesive/structural function in addition to paracellular barrier and polarity functions. There is no attempt in the paper to consider or disentangle these multiple roles. Indeed, the introduction and discussion consider the vertebrate tight junction as the analogue of the insect septate junctions when a better view would be that the septate junction is a combination of the claudin-based barrier function of the vertebrate tight junction and the vertebrate basolateral polarity proteins Scribble, Dlg and Lgl that localize similarly and presumably have a function similar to the *Drosophila* basolateral polarity/SJ proteins for which they are named. Moreover, there are no experiments in the paper to address whether the relevant parameter being sensed in SJ defects is loss of the paracellular barrier, loss of cell adhesion/contact/structure or disruption of the polarity function of the SJ complex. Notably, there aren't any experiments in the paper that test paracellular barrier function. This criticism does not in any way reduce the importance of the paper or the results, but to avoid presenting an overly simplistic and probably misleading view of the cellular processes in play, a more comprehensive discussion of SJs is in order.

The concerns raised by the reviewer are entirely correct. Indeed, in our Ms we were referring the paracellular diffusion barrier function of SJ, that to our mind covers the barrier function to solute diffusion through the intercellular space as well as the structural strength. As such, it gave the impression that we were focusing solely on the paracellular diffusion barrier whereas our study unravels the additional roles of SJ proteins, linked to cell/cell adhesion and structural function as pointed out by the reviewer. We have amended the text, including title, to take this first point into consideration, i.e. replacement of paracellular diffusion barrier by septate junctions that include the two functions, the diffusion barrier and structural function.

We considered the possibility of a cell polarity defect from the outset, but consider that we can rule out this possibility. Indeed, loss of scribble/Dlg causes delocalisation of E-Cad/Arm and of Crumbs to the basal pole, phenotypes we never observed in mutants or upon depletion of SJ components including Aka, M6, Gli, Nrv, Cora. We have added a sentence to explain this in the revised version (line 437).

With regard to the criticism that "*the article contains no experiments testing paracellular barrier function*", our previous work demonstrates the loss of SJ-specific proteins such as NrxIV at the vertices upon loss of TCJ proteins. Here, we are confident that our EM analysis provides irrefutable evidence that Aka depletion leads to the appearance of holes at contacts between three cells in the plane of septate junctions, giving credit for defects in defective permeability barrier function.

line 245: "We propose that it is the Shrub activity that is being modified upon SJ alteration, preventing SJ component degradation in favour of SJ component recycling."line 288, "Thus, as proposed above for Nrx-IV, these data further suggest a hijacking of Shrub activity toward recycling components upon alteration of SJ integrity."Model in Figure 7 Arrows showing increased SJ protein delivery in right bottom panel, but decreased bicellular SJ complex formation in the left bottom panel.The authors demonstrate that in bicellular SJ mutants, there is increased accumulation of Crb, adherens junction components, focal adhesion components, and in the text and in the model in Figure 7 focus on the upregulation of recycling activity. However, as indicated by the reduced bSJs in the left bottom panel in Figure 7, and in the reduced Nrx levels in 3C' and in the text in lines 351-53, the levels of most septate junction proteins drop in the absence of any of 15+ bicellular septate junction mutants. Previously the authors should that reduction of tricellular septate junction proteins increased levels of septate junction proteins in bicellular junctions which the authors translate to increased delivery of "SJ components" to the membrane in SJ mutants as shown in Figure 7 bottom right panel and stated in lines 245 and 288. But the data in the paper, which is consistent with statements on lines 351-353 saying that bicellular SJ mutations cause a general reduction of SJ protein levels, suggests either a more nuanced role of recycling such that Crbs and other proteins show increased recycling in bicellular SJ mutants, but bicellular SJ proteins show decreased recycling, or an alternative scenario in that the SJ proteins are recycled more in a SJ mutant, like Crb is, but SJ proteins don't form stable complexes which leads to their modification that targets them for destruction despite being recycled more. Regardless of the actual explanation, I think readers will be confused by the statements in the current version of the paper about upregulation of recycling activity but apparent reduction of SJ proteins. The authors should address this issue with appropriate changes to text and the model figure.

We thank the reviewer for pointing out to this potential confusion between the description of the data, the model in Figure 7 and the interpretation of the data. We appreciate her/his suggestion of interpretation ‘an alternative scenario in that the SJ proteins are recycled more in a SJ mutant, like Crb is, but SJ proteins don't form stable complexes which leads to their modification that targets them for destruction despite being recycled more.’ We do favour the hypothesis according which not only Crumbs and β-integrin are recycled at higher rates but also SJ components even in the case of loss of a bSJ component. However, in this latter situation, we do not see the bSJ components accumulating at the plasma membrane as they are not forming stable complex (short time residence according to FRAP analyses), nor we detect them in endosomal/recycling compartments. In fact, with the exception of blocking MVB-dependent protein degradation, we only detect SJ components (GFP lines, antibody staining) at the plasma membrane but not in the secretory nor the endocytic compartment, and we have no explanation for this (small compartments, low amounts of protein/ compartment, short residency time?)

Accordingly, we have amended the text (lines 405-410, and 482-485) to better explain our thoughts and taking into account the reviewer’s suggestion.

The assumption in the paper is that the changes in protein levels result from changes in recycling of the proteins. However, it would be nice to rule out transcriptional regulation. Has anyone established smFISH in the notum that would allow quantification of Crb or other marker RNA to show that there is not increased accumulation of the Crb RNA in the SJ mutant backgrounds?

SmFISH is described in whole mounted *Drosophila* brains or embryos, however, we do not master the technique on adult thin tissues. However, in the event of a transcriptional response, this would be a global effect since E-Cad, Crumbs, β-integrin and bSJ components levels are increased upon loss of Aka (M6 or Gli).

In the revised manuscript, we have modified the text (lines 440-445) and argue in favour of the hypothesis of increased recycling, although we cannot exclude an overall effect at transcriptional level.

line 58. SJ are only the functional equivalent of tight junctions for paracellular barrier function. SJ have basolateral polarity function that correspond to basolateral polarity proteins in vertebrates, whereas vertebrate TJs are associated with apical complexes. In addition, the mechanical properties of SJ and TJ are probably wildly different since the SJ is a much more elaborate structure with many more cell-cell adhesion proteins than TJs. I feel the presented over-simplification do not adequately inform the reader about alternative functions and therefore hypotheses about the data in the paper.

We have therefore reworded the paragraph to avoid oversimplification: lines 58-70

lines 120-121 , Figure 1A-A'. Please quantify the relative frequency of holes observed in the EM sections. Is it every tricellular junction or 1 in 100? Is WT statistically different than mutant?

We thank the reviewer for raising this key point. In the submitted manuscript, the section devoted to the EM analysis was missing. We have included it in the revised version, and added the quantitative analyses in Figure 1A’’. We are confident that our analysis allows us to conclude that Aka depletion causes the appearance of these holes, and consequently the loss of barrier function integrity.

Lines 713-737Transmission electron microscopy sample preparationWT and aka RNAi *Drosophila* pupal nota (16h APF) were dissected in 0.1 M cacodylate buffer at room temperature and immediately processed (Kolotuev, 2014). Briefly, the samples were fixed in 1% paraformaldehyde and 2.5% glutaraldehyde in 0.1 M cacodylate buffer for 2h. Then, they were stained for 1h in 2% (wt/vol) osmium tetroxide and 1.5% (wt/vol) K4[Fe(CN)6] in cacodylate buffer followed by 1h in 1% (wt/vol) tannic acid in 100 mM cacodylate buffer. Finally, they were incubated for 30min in 2% (wt/vol) osmium tetroxide followed by 1% (wt/vol) uranyl acetate for 2h. After the dehydration cycles, samples were embedded in Epon–Araldite mix. To ensure precise orientation and access to the samples, a two-step flat-embedding procedure was used (Kolotuev, 2014). Sections were cut with an Ultracut E microtome (Reichert-Jung, Austria, now Leica Microsystems) parallel to the plane of the pupal nota epithelia. Semi-thin sections (0.7–1 μm thick) were mounted on microscope slides and stained with 1% aqueous solution of methylene blue in 1% borax. Ultrathin sections (70–80nm thick) were collected either on standard copper grids or single-slot nickel grids coated with formvar (polyvinyl formal; Polysciences, Inc). The sections were contrasted with saturated aqueous uranyl acetate solution for 20s, rinsed with double-distilled water and stained in Reynolds’ solution (lead citrate; SigmaAldrich) for up to 3min (Reynolds, 1963). After several rinses with deionized water and drying, the sections were examined with a JEM-2100 HT (JEOL Ltd, Japan) transmission electron microscope at 80kV. The pupal nota of three WT and three aka RNAi specimens were examined, and at least 10 ultrathin sections of the region of interest were analyzed in each specimen. At least three tricellular junctions were examined in each set of sections.line 126-127 (data not shown). Does EMBO allow data not shown? Just checking current rules.

We have now added the data on Gli (Figure S1 B,B’) and Coiled (Figure 1G-G’), and we have delete all sentences containing data that are not displayed,

lines 134-135. "We observed similar results upon loss of Gli and M6". Is this data not shown? I couldn't find it. Please either reference a figure or note as "data not shown" if that is allowed.

We have removed the 'data not shown' and instead provided the quantified data set for Gli (Figure S1 B,B') to reinforce the idea that the results were not obtained only when Aka was lost.

line 319 "We propose that the disruption of SJ barrier in the …", also line 326. I suggest the use of "SJ complex" instead of SJ barrier or paracellular diffusion barrier, otherwise the authors need to provide some evidence or rationale that it is the barrier function of the SJ that is triggering the recycling changes rather than the disruption of the polarity or adhesive/structural functions of the SJs.

We thank the reviewer for this suggestion, which we have implemented

line 341 "Our work shows that a part of the sensing mechanism involves the ESCRT machinery."I think that the ESCRT machinery is better described as part of a response mechanism to SJ defects than as a "sensor". I don't think the paper presents any evidence that the ESCRT machinery is part of the sensing mechanism for SJ defects. There is lots of evidence that the ESCRT machinery is modified by SJ defects, but that supports a role as part of the response machinery, not as the sensor that directly detects SJ defects.

We agree with the suggestion; accordingly, we have amended the text, lines 392-397.

Reviewer 2:1. The Abstract states: "We report that the weakening of SJ integrity, caused by the depletion of bi- or tricellular SJ components, reduces ESCRT-III/Vps32/Shrub-dependent degradation and promotes instead Retromer-dependent recycling of SJ components." This is too strong, as the role of the retromer, while plausible, is not directly tested. It's fine to speculate about this in the Discussion but drawing a conclusion like this in the Abstract is unwarranted.

We thank the reviewer for pointing this out. We agree with her/him and have made the appropriate changes in the abstract.

2. Similarly, the title suggests that "ESCRT-III-dependent adhesive and mechanical changes are triggered by a mechanism sensing paracellular diffusion barrier alteration". They show that knocking down septate junctions alters localization of vesicle trafficking machinery, and that it leads to alterations in apparent recycling of cargo, but do they ever really assess whether these changes are ESCRT-III-dependent? Wouldn't this require knocking down ESCRT-III in cells with defects in septate junctions? There was a lot of data in this paper and perhaps I missed it but was this experiment done? I am not suggesting they do it, but that they temper this conclusion if not.

We have generated a line to simultaneously deplete Cora and Shrub. This turned out to be lethal too early to be analysed.

Therefore, to take into consideration the reviewer’s comment, we have toned down our statements about ‘ESCRT-III dependency throughout the revised manuscript’.

3. The authors assessed "poly-ubiquitinylated proteins aggregates appearance, marked using anti-FK2". They need to define FK2-what does it detect.

We thank the reviewer for having noted this error, and we have amended the text to explain what the anti-FK2 is and what it recognises, lines 265-266, and 977.

4. Figure 4-is this a clone, and are we far from the boundary? Make this clearer

The data in Figure 4 A,C and D are not clones, here we have used *scabrous-* or *pannier*-Gal4 drivers to drive the RNAi of Shrub (A), Cora (C) or Nrx-IV (D). The *sca*-depleted area is validated by the co-expression of a GFP probe (UAS-KAEDE) with the RNAi.

Pannier is expressed in the central part of the notum (panel C and D) that is easily detected by morphological means (midline of the animal). The control area (Figure 4 B) corresponds to lateral portion of the notum. In addition to the morphological means, immunolabelling of the targeted SJ component (Nrx-IV) demonstrates the efficacy of RNAi-based depletion as well as the area of tissue analysed.

5. The authors state: "Despite these apparent similarities, we noticed that, in contrast to Shrub depletion, NrxIV did not accumulate in enlarged intracellular compartments upon Cora depletion"Could the authors reference a Figure here?

We have amended the text accordingly, lines 277-279.

Despite these apparent similarities, we noticed that, in contrast to Shrub depletion (Bruelle et al., 2023), NrxIV did not accumulate in enlarged intracellular compartments upon Cora depletion (Figure 4C and 4C’’’).6. The authors state: "Hence, if both Shrub and bSJ/tSJ defects lead to Crumb enhanced signals" It might be better to say "altered" as they then point out the differences.

We have amended the text accordingly, line 314.

7. I found the Discussion challenging to follow. Rather than focusing on the core observations, it addresses many, not very well-connected speculative possibilities, and in my opinion, will be challenging for most readers to follow. I would encourage the authors to revisit it from top-to-bottom.

It is difficult for us to respond specifically to this general comment on the discussion. We have reworded the discussion to consider the criticisms made by the three reviewers and we tried to better connect the hypotheses formulated.

We have shortened and simplified the section on ‘cell mechanics and adhesion’ and added a section on the potential conservation of the process.

Reviewer 3:The title refers to a "mechanism sensing paracellular diffusion barrier alteration", and in the discussion (line 325) authors state that "loss of bSJs and tSJs by altering the paracellular diffusion barrier triggers an ESCRT-dependent response…". However, no experiments to assess paracellular barrier function (epithelial permeability) are shown in the paper, and it is not clear that the ESCRT-dependent responses described here are triggered by altered barrier function per se, as stated by the authors, or by changes in other SJ-dependent parameters, such as cell adhesion or intra-membrane mobility of lipids and proteins. Statements about paracellular barrier alteration should be rephrased accordingly.

We thank the reviewer for raising these issues, which were also raised by reviewer 1. In accordance, we have rephrased the sentences (see also our response to reviewer 1).

Altered epithelial barrier function will likely influence osmoregulation via changes in organismal hormonal status and gene expression, which may contribute to the phenotypes described here. How much time passed between induction of mutant clones and phenotypic analysis? The authors should discuss these aspects, and consider that effects of altered barrier function will depend on the distribution and size of clones with defective SJs.

In this study, we have analysed clones of mutant cells at 16h after formation of the pupae. Heat shock to induce clones were done at L2 and L3 stages. The effect of gene silencing on the pupal notum (using pnr and sca-Gal4 drivers) is initiated at L3 stage. The phenotypical consequences are therefore analysed 2 to 4 days after the induction of clones or gene silencing. In that sense, the reviewer is fully correct and we cannot exclude hormonal and gene expression defects at the organismal level.

However, we have systematically observed the effects reported on E-Cad, Crb, Integrin, Shrb, FK2, etc. independently of clone position on the notum and clone size. In addition, these effects remain cell-autonomous. We do not expect this to be the case in the event of a defect (hormonal or gene expression) at the organism level.

In the discussion the authors speculate about a "sensing" mechanism based on (hypothetical) altered membrane lipid composition upon loss of SJs. However, such effects would not explain how altered barrier function per se (epithelial permeability) would be sensed by cells, as stated in the title and throughout the text. Please explain.

The reviewer is right, the point we are making here is to say how an SJ defect can lead to a lipid composition defect which in turn would lead to a transport defect (as shown for Mfsd2 for BBB in vertebrates). This argument provides an explanation of how this could contribute to the transport defect, but in no case a detection mechanism. We have amended the text and reformulated it as such in the revised version.

How Shrb/ESCRTIII activity could be "redirected" or "modulated" by disruption of SJs remains unclear. Can the authors briefly outline possible mechanisms for modulation of ESCRT activity?

As discussed in the manuscript, recent work reports that ESCRT function is required for apical localization and mobility of retromer positive carrier vesicles (Pannen et al., 2020). One could envisage that upon loss of Aka or bSJ component, a specific subset of ESCRT cargoes (perhaps SJ components themselves) are found in higher amounts in endosomes and favors the above mentioned function of ESCRT instead of promoting MVB formation and ubiquitinated protein degradation.

A second possible explanation relies on a repair mechanism. ESCRTIII is known to be recruited to the plasma membrane or to the membrane of organelles in the event of injury in order to repair it. This mechanism relies on the detection of calcium. It is therefore conceivable that a change in the composition of the SJ could send a signal via calcium, leading to ESCRTIII being 'hijacked' to these locations. As the SJ is continuously perturbed in our situation, ESCRTIII is continuously addressed to the cell membrane.

In plants (https://www.ncbi.nlm.nih.gov/pmc/articles/PMC9171914/), ESCRT machinery (ALiX) interacts with the retromer core subunits, by recruiting Retromer onto endosomes to trigger recycling of Vacuolar sorting receptors. By analogy to plant, a third possibility would be that increased levels of ESCRT on endosomes could result in increased recruitment of retromer to trigger recycling of cargos (including Crb, β integrin, SJ components). These possibilities are now discussed in the revised Ms.

The presentation of fluorescence intensity data in a rescaled ("standardized") format is uncommon and non-intuitive, as it obscures the true scale (fold-changes) and variation of the data. Also, if data were plotted as a range from 0 to 10, as stated in Materials & Methods, it is not clear why in all graphs (except for a single datapoint in Figure 5C'?) values start at 1, not at 0. Highest values appear to cluster at 10 and lowest values at 1, suggesting these represent saturated or clipped signals, respectively. Were these datapoints taken into consideration for calculating mean values? Authors need to explain exactly how the analysis was done. Why was this type of representation chosen, and why should it be more appropriate than showing regular normalized data?

We thank the reviewer for her/his comment. We mistakenly used the term standardisation instead of normalisation. The min-max normalisation between 1 and 10 was done as described here: Normalized value=1+(Value − Min Value)x(10−1)(Max Value − Min Value). By doing so, 1 correspond to the lowest value of the dataset and 10 the maximum.

For the graph in Figure 5C’, there was a mistake in the data analysis (error in the selection of the minimal value in two datasets), we apologize for this error and thank the reviewer for pointing it out.

We have made the appropriate changes in the revised manuscript.

Authors should explain why they jump between different mutant (aka, nrv2) and RNAi (aka, cora nrv2, nrxIV) conditions and different Gal4 driver lines (pnr-Gal4, sca-Gal4) to disrupt SJ integrity. The basis for choosing these different conditions is not always clear and makes results difficult to compare.

Whenever possible, we diversified tools to confirm that what we observed in mutant conditions were also observed using another independent approach. We used Cora and NrxIV RNAi due to the limited genetic tools to induce mosaic clones for those genes. We choose Sca over Pnr when it was too detrimental for the pupa. For instance, pnr-Gal4 > Shrub RNAi does not allow us to analyse pupae as it dies before reaching 16h APF. In that case, we had to use Sca-Gal4 which allows pupal development.

Explaining all these subtleties and experimental limitations seems to us to make the text even more cumbersome, so we have decided not to provide these details systematically.

In the revised Ms, we will specify the reasons for which this or that tool is used.

The TEM images shown in Figure 1A are difficult to interpret, because plasma membrane is barely visible. The images do not seem to contribute much and can be removed from the paper.

The reviewer is correct by saying that the plasma membranes are not always easily detectable. To be more precise, they are not nicely delineated along their entire length and show low contrast. Also, one should keep in mind that the sections are only 70-80 nm thick and the appearance of the cell membranes very much depends on a sectioning plane. Nevertheless, we can make out the separation between neighboring cells of the notum (indicated by arrows in the micrographs).

We believe this EM dataset of epithelial sections parallel to the plane of the epithelium is essential to demonstrate the holes at TCJ resulting from aka depletion, a phenotype never observed in the control situation (see also our answer to reviewer 1 concerning the quantitation of the phenotype).

This is why we have retained this essential data to demonstrate the loss of integrity of the epithelial barrier for the revised version.

The position of mutant clones is marked by absence of nuclear RFP (Figure 1B and elsewhere), but drawings of clone boundaries (Figure 1B) do not match with the pattern of RFPpositive/ -negative nuclei (Figure 1B'), presumably because different optical sections are shown in Figure 1B and B'. This is confusing and needs to be explained.

This is indeed due to different optical sections. To ascertain the clone boundary in the apical plane, we draw boundaries at the apical level by maximising the nls-RFP signal.

In the revised version, we have modified the Figures as follows:

Figure S1 A-A" explains how clonal boundaries are unambiguously determined by showing apical and basal confocal sections.

In Figure 1A-C, E-G, and following, we have removed the panels showing the nls::RFP marker that was used to determine clone boundaries and instead have labelled clone boundaries only. This makes the data clearer and avoids the reader wondering how the clone boundaries were defined.

Line 102: "We recently reported that defects at tricellular Septate Junctions (tSJs) are always accompanied by bicellular Septate Junctions (bSJs) defects". Authors may want to mention that in embryonic and larval epithelia lacking tricellular SJs, bicellular SJs assemble initially, but appear to degenerate during later development (Hildebrand et al. 2015, Byri et al. 2015).

We have made made the appropriate changes, line 124.

Line 192 remove "another".

Done

Line 194: % enrichment and fold enrichment are used; stick to one way.

We have made the appropriate changes.

Line 259 and elsewhere: Crb "activation" vs. accumulation or mislocalization. What do the authors mean by Crb "activation"?

Unless we are mistaken, line 259 mentions ‘Crb activity’ (not activation), which refers to the article (Dong et al., 2014) i.e. Crb presence at apical level triggers apical membrane overgrowth thanks to Crb signalling.

In the sentences, where we used the term "activation", it has been replaced by "activity".

Line 346: "FK2 protein": the FK2 antibody does not detect a particular protein, but the polyubiquitin modification, presumably on many different proteins.

We have corrected this mistake (See also point 3 Reviewer 1)

Line 444: "Also, the observed changes at apical level might be mostly due to direct effects." I don´t see experimental evidence to support that the observed changes are mostly due to direct effects. Rephrase or remove.

We agree with the reviewer, we have amended the text and removed this sentence from the last paragraph.

Information on how mutant clones were induced needs to be included in Materials and methods.

The information has been included as follows (lines 749-754)

Results referred to as "not shown" should be shown, or corresponding statements be removed from the paper

We have made the appropriate changes (see also our response to reviewer 1)

The text needs to be carefully checked for grammatical and typographical errors

We have carried out a critical proofreading and tried as far as possible to check the grammar and correct typos.

[Editors’ note: what follows is the authors’ response to the second round of review.]

Reviewer #3 (Recommendations for the authors):The authors have addressed most of my comments.The issue about the TEM images remains, as "cell membrane detachment" cannot be seen clearly on the image in Figure 1A'. The authors should rephrase their following conclusions accordingly:Line 143: "… we report that, depletion of Aka induces weaknesses in the integrity of the tissue which results in cell membrane detachment at the vertex in the plane of the SJ with the formation of sizeable intercellular gaps within the epithelium (Figure 1A-A')."

Accordingly, we have rephrased this sentence as follows to tune down the notion of gaps:“we report that, depletion of Aka induces weaknesses in tissue integrity manifested by the appearance of sizeable intercellular holes at presumptive TCJ in the plane of SJs (Figure 1A–A’’).“

and included

“These observations are reminiscent to the paracellular cavities observed in embryos lacking Aka or Gli, interpreted as being due to a loss of cell-cell adhesion (Byri et al., 2015; Hildebrandt et al., 2015; Schulte et al., 2003). To investigate whether this morphological defect affects overall epithelial integrity,…”

Line 390: "… weakening the three-cells contact ultimately leading to gaps [this study)…".Also, authors should mention that plasma membrane-lined delaminations in aka or in Gli mutant epithelia were described previously (Schulte et al. J Cell Biol 2003; Hildebrandt et al. Dev Biol 2015; Byri et al. Dev Cell 2015).

We amended the sentence:

“…weakening the three-cells contact as suggested by the holes observed by TEM [this study], presumably preventing the cells from fulfilling their paracellular diffusion barrier function.”

The discussion still contains rather extensive speculation, which should be marked as such, e.g., the subheading on line 387 ("A mechanism detecting bicellular septate junction defects"). Given the largely speculative nature of the arguments discussed here, the authors should consider changing the wording, e.g., to "How could septate junction defects be detected?".

We agree with the Reviewer and have amended the subheading as suggested:

"How could septate junction defects be detected?".

It is not clear to me how the fact that SJ components include GPI-anchored proteins would support the idea that (line 428) "the lipid composition of the lateral plasma membrane is likely to be affected upon the loss of SJ components.". Please either explain or remove this point.

We reformulated this sentence in the form of a proposal, as follows:

“…components are GPI-anchored proteins and that, for example, *wunen-1* and *wunen-2* encode lipid phosphate phosphatase (Ile et al., 2012), raising the question of whether the lipid composition of the lateral plasma membrane can be altered by the loss of SJ components.”